# Exome and genome sequencing of nasopharynx cancer identifies NF-κB pathway activating mutations

Yvonne Y. Li[1],*, Grace T.Y. Chung[2],*, Vivian W.Y. Lui[3,4],*, Ka-Fai To[2], Brigette B.Y. Ma[5], Chit Chow[2], John K.S. Woo[6], Kevin Y. Yip[7], Jeongsun Seo[8], Edwin P. Hui[5], Michael K.F. Mak[2], Maria Rusan[1], Nicole G. Chau[1], Yvonne Y.Y. Or[2], Marcus H.N. Law[2], Peggy P.Y. Law[2], Zoey W.Y. Liu[2], Hoi-Lam Ngan[4], Pok-Man Hau[2], Krista R. Verhoeft[4], Peony H.Y. Poon[4], Seong-Keun Yoo[8], Jong-Yeon Shin[8], Sau-Dan Lee[7], Samantha W.M. Lun[2], Lin Jia[9], Anthony W.H. Chan[2], Jason Y.K. Chan[6], Paul B.S. Lai[10], Choi-Yi Fung[4], Suet-Ting Hung[4], Lin Wang[11], Ann Margaret V. Chang[12], Simion I. Chiosea[11], Matthew L. Hedberg[13], Sai-Wah Tsao[9], Andrew C. van Hasselt[6], Anthony T.C. Chan[5], Jennifer R. Grandis[14], Peter S. Hammerman[1] & Kwok-Wai Lo[2]

Nasopharyngeal carcinoma (NPC) is an aggressive head and neck cancer characterized by Epstein-Barr virus (EBV) infection and dense lymphocyte infiltration. The scarcity of NPC genomic data hinders the understanding of NPC biology, disease progression and rational therapy design. Here we performed whole-exome sequencing (WES) on 111 micro-dissected EBV-positive NPCs, with 15 cases subjected to further whole-genome sequencing (WGS), to determine its mutational landscape. We identified enrichment for genomic aberrations of multiple negative regulators of the NF-κB pathway, including *CYLD*, *TRAF3*, *NFKBIA* and *NLRC5*, in a total of 41% of cases. Functional analysis confirmed inactivating *CYLD* mutations as drivers for NPC cell growth. The EBV oncoprotein latent membrane protein 1 (LMP1) functions to constitutively activate NF-κB signalling, and we observed mutual exclusivity among tumours with somatic NF-κB pathway aberrations and LMP1-overexpression, suggesting that NF-κB activation is selected for by both somatic and viral events during NPC pathogenesis.

[1] Department of Medical Oncology, Dana-Farber Cancer Institute, Harvard Medical School, Boston, Massachusetts, Cancer Program, Broad Institute of Harvard and MIT, Cambridge, Massachusetts 02142, USA. [2] Department of Anatomical & Cellular Pathology, State Key Laboratory in Oncology in South China and Li Ka Shing Institute of Health Science, The Chinese University of Hong Kong, Hong Kong, China. [3] School of Biomedical Sciences, Faculty of Medicine, The Chinese University of Hong Kong, Hong Kong, China. [4] Department of Pharmacology and Pharmacy, School of Biomedical Sciences, Li-Ka Shing Faculty of Medicine, The University of Hong Kong, Hong Kong, China. [5] State Key Laboratory of Oncology in South China, Sir Y.K. Pao Centre for Cancer, Department of Clinical Oncology, The Chinese University of Hong Kong, Hong Kong, China. [6] Department of Otorhinolaryngology, Head and Neck Surgery, Prince of Wales Hospital, The Chinese University of Hong Kong, Hong Kong, China. [7] Department of Computer Science and Engineering, The Chinese University of Hong Kong, Hong Kong, China. [8] Genomic Medicine Institute, Medical Research Center, Seoul National University, Seoul 10-799, Republic of Korea. [9] School of Biomedical Sciences and Center for Cancer Research, Li Ka Shing Faculty of Medicine, The University of Hong Kong, Hong Kong, China. [10] Department of Surgery, Prince of Wales Hospital, The Chinese University of Hong Kong, Hong Kong, China. [11] Department of Pathology, University of Pittsburgh Medical Center, Pittsburgh, Pennsylvania 15261, USA. [12] Institute of Pathology, St. Luke's Medical Center, Quezon City 1112, Philippines. [13] Medical Scientist Training Program, University of Pittsburgh–Carnegie Mellon University, Pittsburgh, Pennsylvania 15261, USA. [14] Department of Otolaryngology, University of California San Francisco, San Francisco, California 94113, USA. * These authors contributed equally to this work. Correspondence and requests for materials should be addressed to P.S.H. (email: peter_hammerman@dfci.harvard.edu) or to K.-W.L. (email: kwlo@cuhk.edu.hk).

N asopharyngeal carcinoma (NPC) is one of the most aggressive head and neck cancers and frequently metastasizes to distant lymph nodes and organs[1,2]. Its distinct etiology linked with latent infection of the oncogenic Epstein-Barr virus (EBV) and characteristic heavy lymphocyte infiltration often distinguish NPC from other head and neck cancers. In fact, EBV-associated NPC is endemic in Southern China and Southeast Asia, contrasting with the epidemic Human Papillomavirus (HPV)-associated head and neck cancer in some Western countries. Due to the intrinsic invasiveness and asymptomatic nature of the disease, majority of NPC patients are diagnosed with advanced diseases (60–70% cases) with poor outcome. Thus far, there is no effective targeted therapy for advanced NPC.

NPC is a complex malignancy with etiology and pathology involving both EBV infection and a combination of multiple genetic aberrations. Unlike other head and neck cancers, the scarcity of NPC genomic data to date hinders the understanding of NPC biology, disease progression and rational therapy design for better treatments. The extensive stromal components of these often small-sized tumours present major challenges for comprehensive and precise genomic characterization.

Here we report the genomic landscape of the first and largest cohort of microdissected EBV-positive NPC of more than 100 cases. Our study suggests that the somatic mutation rate of NPC is substantially higher than that reported previously[3] and reveals that majority of NPCs display activation of the NF-κB signalling pathway as a result of somatic inactivating mutations in negative regulators of NF-κB.

## Results

### Whole-exome sequencing (WES) of microdissected NPC.
WES was performed on 111 unique tumour specimens derived from 105 unique subjects. The cohort included 78 primary tumours, 11 local recurrences and 22 metastatic tumours. Ninety-seven subjects were from Southeast Asia where NPC is endemic and eight from the United States (Supplementary Table 1 and Supplementary Data 1). The overall non-synonymous mutation rate of EBV-positive NPC was significantly lower than that of HPV-positive head and neck squamous cell carcinoma (HNSCC)[4] and EBV-positive stomach adenocarcinoma[1] ($P = 0.0196$ and 0.0004, respectively; unpaired $t$-test; Fig. 1a). Primary, recurrent and metastatic NPC tumours shared a similar mutational burden in our cohort and orthogonal validation of mutations by amplicon sequencing demonstrated a validation rate of 82% at selected sites. Interestingly, increasing mutational burden in primary NPC lesions ($N = 70$) was negatively correlated with overall and disease-free survival ($P = 0.0214$ and 0.0118, respectively; Log-rank test; Fig. 1b). NPC mutational rate was not associated with age, sex or clinical stage.

The combination of EBV infection, diet and genetic factors are etiologically linked with NPC pathogenesis in endemic regions[5].

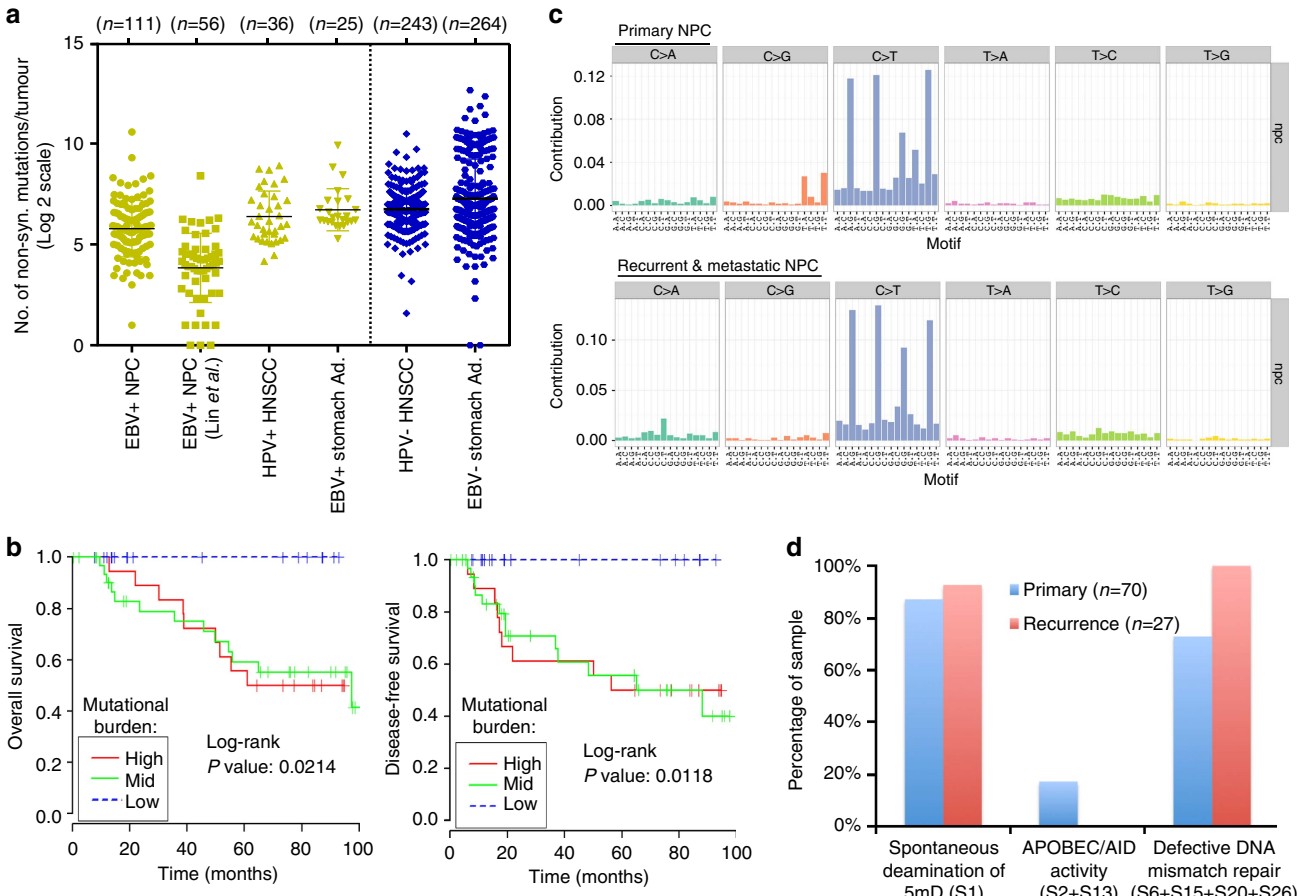

**Figure 1 | Somatic mutation rates and mutational signatures of NPC tumours identified by WES.** (**a**) NPC tumours have significantly lower rate of non-synonymous mutations as compared with HPV-positive HNSCC and EBV-positive stomach adenocarcinoma (TCGA, USA). (**b**) The mutational burden, stratified by the top, mid and low quartile counts per patient, correlated with overall and disease-free survival in NPC patients ($N = 70$ NPC patients with primary tumours). (**c**) Mutation signatures of 70 primary and 27 recurrent or metastatic NPC cases reveal a primarily C to T transition signature. (**d**) The top COSMIC cancer mutation signatures in NPC.

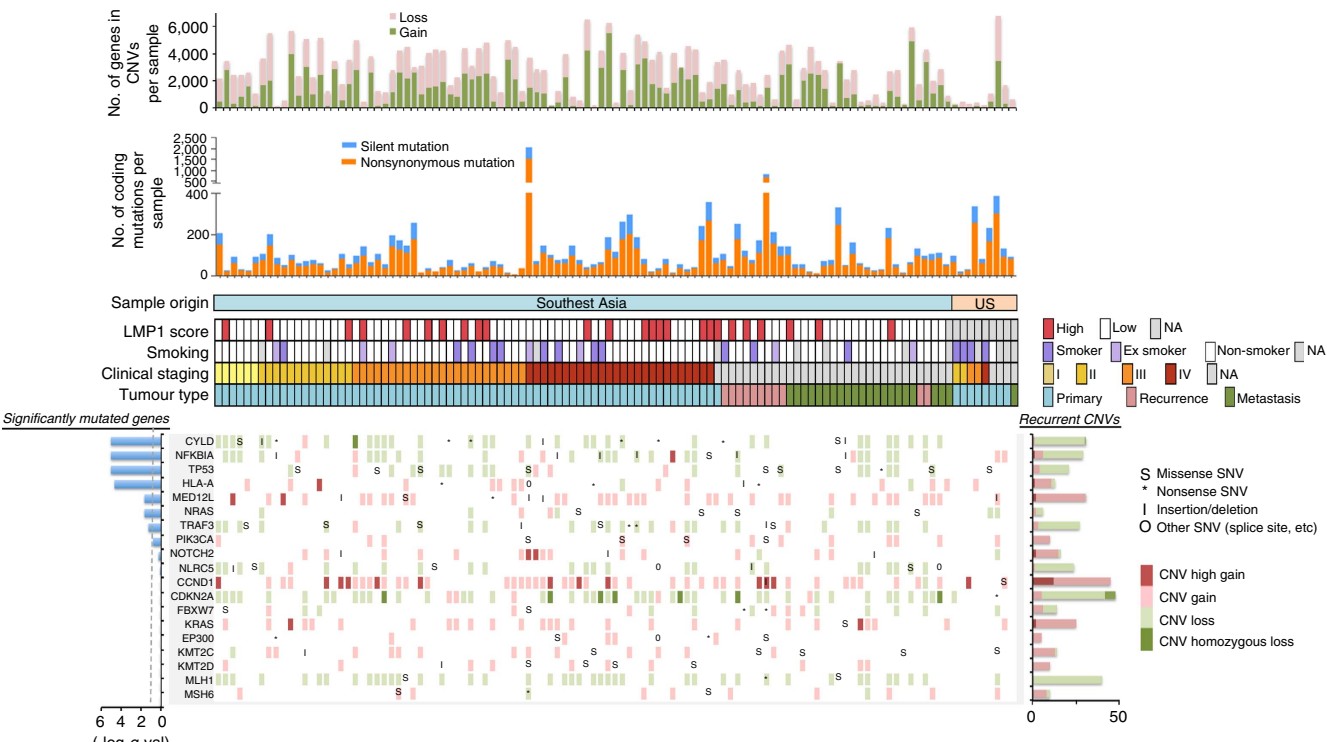

**Figure 2 | The genomic landscape of NPC.** For each patient (each column), recurrently altered genes (rows) with mutations and CNVs are shown. Significantly mutated genes (identified using the MutSigCV 2.0 algorithm; $q < 0.1$, left panel) are ordered by $q$ value, with additional genes near significance are also shown. Recurrent gene copy changes across the patients are summarized on the right panel. For each patient, the number of mutations and CNVs are shown on the top panels, as well as the sample origin, latent-membrane protein 1 (LMP1) expression, smoking status, clinical staging and tumour type.

To identify whether mutagenic processes are operative in NPC, we analysed the mutation spectrum of NPC. The predominant type of substitution was C > T transition at NpCpG sites in both primary and recurrent/metastatic tumours (Fig. 1c). Combined non-negative matrix factorization clustering and correlation with the 30 curated mutational signatures defined by the Catalog of Somatic Mutations in Cancer (COSMIC) database[6,7] revealed three dominant signatures in NPC (Supplementary Data 2). The predominant signature underlies the deamination of 5-methyl-cytosine process (Signature 1), followed by defective DNA mismatch repair signature (Signatures 6 + 15 + 20 + 26), and by an APOBEC/AID signature, which was only observed in a subset of primary NPCs (Signatures 2 + 13) (Fig. 1d).

**Negative regulators of NF-κB pathway are commonly mutated.** To identify somatically mutated genes in NPC, we employed the MutsigCV 2.0 algorithm[8] with a Benjamini-Hochberg false discovery (FDR) threshold of 0.1 (Figs 2 and 3a). We identified a total of seven genes significantly mutated above background rates in Asian NPCs: TP53, NRAS, NF-κB pathway genes, CYLD, TRAF3 and NFKBIA, the major histocompatibility complex (MHC) class I gene HLA-A, and the transcriptional regulator MED12L, a gene previously reported to be mutated in mouse and human HNSCC[9] (Supplementary Data 3 and 4). TP53 mutations were observed in 9.5% of cases, consistent with rates reported previously[1,4]. Yet, TP53 mutations were ~2.3-fold enriched in recurrent/metastatic NPC (15.2%; versus 6.4% in primary NPC), perhaps due to therapy-related selection. TP53 mutation status was not correlated with survival of patients with primary NPC. Our US cohort size was not large enough to enable statistically robust comparisons with the Asian cohort of significantly mutated genes.

CYLD, TRAF3 and NFKBIA are important negative regulators of NF-κB activity. CYLD is a tumour suppressor gene that deubiquitinates TRAF2, an activator of NF-κB signalling[10,11]. Germline mutations in CYLD are associated with familial cylindromatosis, characterized by multiple benign tumours in the head and neck region and hair follicles[12]. We found that the majority of CYLD mutations (8/11 mutations; 72.7%) were truncating, including nonsense and frameshift mutations. Truncating mutations of CYLD have not been identified in NPC previously, but were found at lower rates in HNSCC[4]. TRAF3, a key negative regulator of the non-canonical NF-κB pathway in NPC, was found to be mutated in 8.6% cases (9/105 cases), a rate higher than previously reported in NPC (1/33 cases; 3.0%) (refs 13,14). Most NPC-associated TRAF3 mutations were located in the RING finger and the MATH/TRAF domains (Fig. 3a), regions known to be essential for the suppression of NIK-activating NF-κB signalling[13]. Thus, mutations at these sites may lead to constitutive activation of NF-κB signals in NPC tumours. NFKBIA functions to trap NF-κB in the cytosol, thus inhibiting its activation, and was mutated in 6.7% of NPC. Deletion of this gene has been reported in glioblastoma, which was associated with poor survival[15].

Additional genes which displayed recurrent mutations in our data set but did not reach statistical significance ($q < 0.1$) included the NF-κB pathway member NLRC5, which was mutated in six cases (4.8%) (Fig. 3a). NLRC5 is a potent inhibitor of NF-κB activation and Type I interferon signalling[16]. It competes with NEMO for binding to IKK-alpha and IKK-beta, thus blocking their activation and kinase activities. Notably, NLRC5, is also a key transcriptional activator of MHC class I genes, including HLA-A and HLA-B. In fact, HLA-A was significantly mutated in our cohort (7.2% of cases), with predominantly recurrent truncating mutations. These findings implies that inactivation

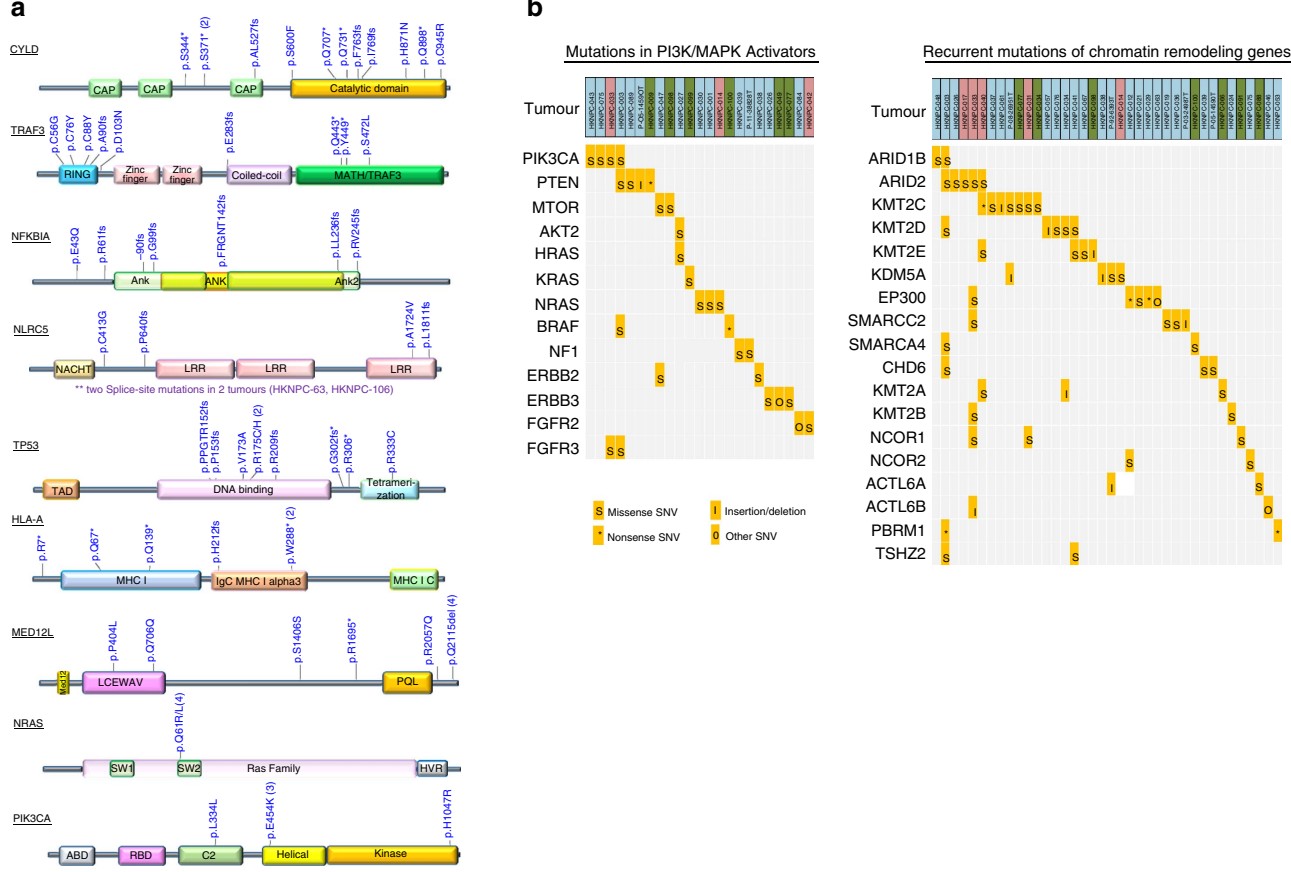

**Figure 3 | Recurrent somatic mutations in NPC.** (**a**) Mapping of mutation sites of *CYLD, TRAF3, NFKBIA, NLRC5, TP53, HLA-A, MED12L, NRAS* and *PIK3CA* from this NPC cohort. Functional domains of the altered proteins are based on UniProt database. (**b**) Non-synonymous mutations in chromatin remodelling genes and PI3K/MAPK activating genes in NPC. Primary NPCs are shown in light blue boxes, local recurrent and metastatic tumours in pink and green boxes, respectively.

of MHC class I molecules may have a role in the pathogenesis of a subgroup of NPC.

**Mutations in PI3K/MAPK pathways and chromatin modification.** Recurrent mutations in PI3K/MAPK pathway activators or regulators (*PIK3CA, PTEN, ERBB3, BRAF1, NF1, FGFR2, FGFR3*) and chromatin modifying enzymes (*KMT2D, KMT2C, EP300, KDM5A*) were also observed, but infrequently, in agreement with a previous study[3] (Fig. 3a,b, Supplementary Fig. 1). Interestingly, recurrent mutations of genes previously reported to be mutated in cancer: *ABL1, BUB1B, NCOR1* (each mutated in three tumours), and *CARS, HSP90AB1, NCOA1* (each mutated in two tumours) were uniquely found in recurrent/metastatic NPCs (*n* = 33). Among these, *ABL1* K690del was identified in a NPC liver metastasis (HKNPC-091) and a lymph node metastasis (P-10-27817). This *ABL1* mutation has been identified in melanoma by the TCGA (www.cbioportal.org). We identified two individual's tumours with hypermutator phenotypes (HKNPC-003, HKNPC-033) with 2,051 and 817 somatic mutations in coding regions, respectively (Fig. 2); both tumours harboured mutations of mismatch repair genes, including *MSH6* (HKNPC-003) and *MLH1* (HKNPC-033). Both patients with hypermutated tumours had poor overall survival of ∼12 and 19 months, respectively.

**Clonal expansion in NPC recurrences and metastases.** Multiple tumours were available from four patients for clonality analyses by comparing variant frequencies between the primary and

respective recurrent/metastatic lesions from the same individual (Supplementary Fig. 2, Supplementary Data 5). Surprisingly, we noted that many mutations with previously annotated roles in cancer were not shared between the paired primary and recurrent lesions. This observation from our two primary/recurrent tumour pairs (cases HKNPC001 and HKNPC012) seems to concur with our recent findings in primary/recurrent HNSCC. In paired primary/recurrent HNSCC samples, we also found that *TP53* was the only shared annotated mutated cancer gene as defined by the Cancer Gene Census, COSMIC[17]. For these NPC patients, the recurrent or metastatic lesion appeared to show a single-nucleotide variant (SNV) clonality distribution different from that of the primary tumour, supportive of emerging new subclone(s) in recurrences. On the other hand, the liver metastases from patients HKNPC-008 and HKNPC-009 both showed a relatively higher percentage of shared mutations among two distinct regions of liver metastases (Supplementary Fig. 3; Supplementary Data 5).

**Frequent chromosomal deletions of Chr 3p, 14q and 16q.** Regions of somatic copy number alteration were defined using WES and whole-genome sequencing (WGS) segmentation data and the GISTIC tool[18] (Fig. 4; Supplementary Figs 4 and 5). Similar gene copy changes were observed in our WES (from formalin-fixed paraffin-embedded (FFPE) tissues) and WGS (from fresh frozen tissues) studies. Selected copy number variants (CNVs) were further validated and confirmed by

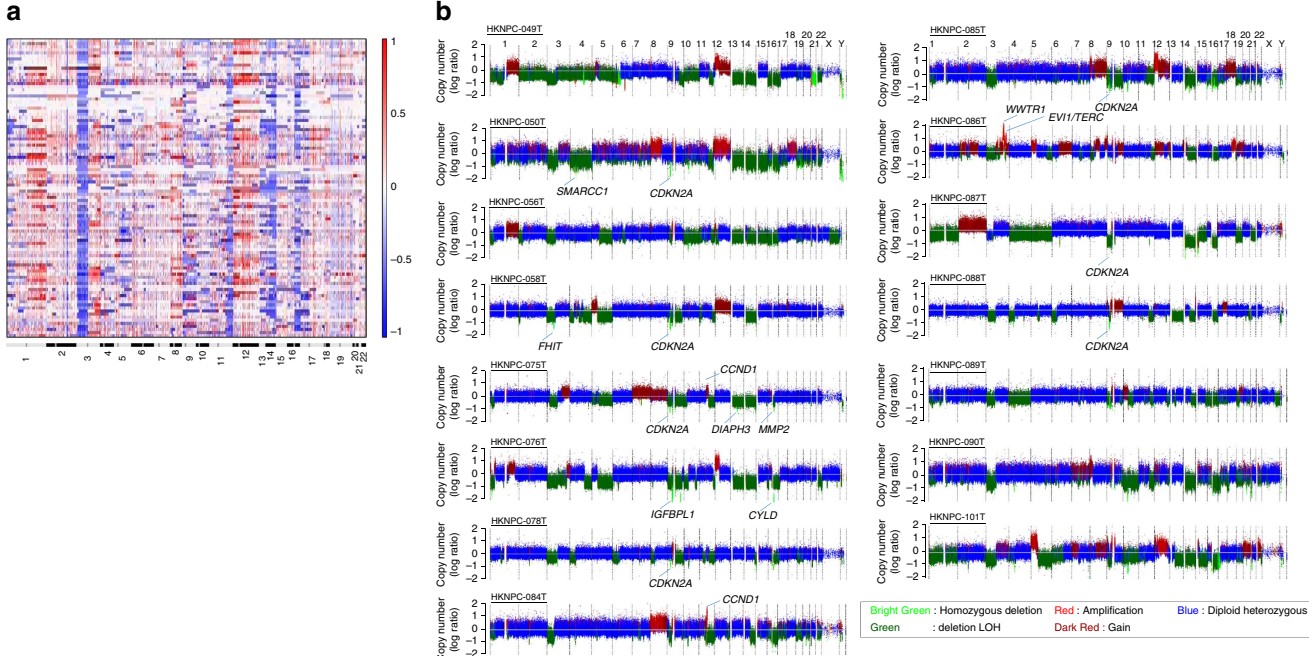

**Figure 4 | Copy number alterations in NPC.** (**a**) By WES, global chromosomal gains (shown in red) and losses (shown in blue) across 97 NPC exomes from unique Asian NPC patients (rows) show recurrent arm-level CNV events. (**b**) Genome-wide view of copy number aberrations in 15 NPC tumours with WGS. Homozygous deletion regions containing cancer genes such as *CDKN2A, FHIT, SMARCC1, CYLD* are indicated. Amplification of 11q13 containing *CCND1* is shown in HKNPC-075T and HKNPC-084T.

fluorescence *in situ* hybridization (FISH) analyses (Supplementary Fig. 6). Arm-level copy number analysis demonstrated consistent losses of 3p, 14q, 16q and recurrent 12p amplification. As we reported previously, 3p loss contributes to the inactivation of multiple NPC-related tumour suppressors including *RASSF1A* and *FHIT*[3,19]. Losses of 14q and 16q may serve as one of the mechanisms for inactivating multiple negative regulators of NF-κB pathway, such as *NFKBIA* (14q13), *TRAF3* (14q32.3), *CYLD* (16q12.1) and *NLRC5* (16q13). Notably, deletion of *CYLD* was identified in 20/97 cases (20.6%) with one case harbouring homozygous deletion (Supplementary Fig. 5). The *CYLD* homozygous deletion was further confirmed by FISH (Supplementary Fig. 7). Other recurrent focal copy number peaks included loss of the *CDKN2A/CDKN2B* locus (9p21.3; 57.6%), consistent with our previous findings[20].

**CYLD is tumour-suppressive in NPC.** To further define the scope of genetic alterations in NF-κB signalling at the whole-genome level, we determined the structural variants from 15 whole-genome sequenced Asian NPC tumours (Supplementary Data 6). We also performed FISH analyses for *CYLD, TRAF3 and NFKBIA* in all Asian NPCs. For both *CYLD* and *TRAF3*, inactivating structural alterations, including translocation, tandem duplication and homozygous deletion, were identified in seven additional cases (Fig. 5a,b; Supplementary Figs 7 and 8). No additional cases with *NFKBIA* structural alteration were identified. The summative rate of somatic genetic events in *CYLD* (18.6%) and *TRAF3* (17.5%) was as high as ~35.1% (with one case having both *CYLD* and *TRAF3* alterations), suggesting that *CYLD* and *TRAF3* genomic aberrations may represent a major mechanism underlying NPC tumorigenesis.

We and others have previously reported cellular studies of *TRAF3* in NPC models[14]. Here we further showed that wild type *TRAF3*, but not the patient-derived mutants inhibited the non-canonical NF-κB pathway in NPC cells (Supplementary

Fig. 9). On the other hand, little is known about the functional importance of *CYLD* mutations in NPC or other cancers. Importantly, a *CYLD* frameshift mutation was also identified in the EBV-positive NPC cell line, C666-1, resulting in complete loss of CYLD protein expression in this cell model (Supplementary Fig. 10). Ectopic expression of *CYLD* wild-type gene suppressed C666-1 cell proliferation (Fig. 5c) and anchorage-independent growth on soft agar (Fig. 5d), while expression of patient-derived *CYLD* point and truncating mutations resulted in the loss of these tumour-suppressive activities (Fig. 5c,d). This loss-of-function activity of *CYLD* mutants was confirmed in another NPC cell line, HK1-EBV (Supplementary Fig. 11). Further, a higher NF-κB transcriptional activity (as measured by an NF-κB-luciferase reporter assay) was observed in C666-1 cells expressing the patient-derived *CYLD* mutants versus wild-type (Fig. 5e). Furthermore, nuclear translocation of Bcl-3, a component of reported NPC-associated activated NF-κB signal p50/p50/Bcl-3 (ref. 14), was inhibited upon *CYLD* WT expression, but not by *CYLD* mutants (Supplementary Fig. 12).

**LMP1 and NF-κB pathway aberrations are mutually exclusive.** The viral oncoprotein, LMP1 is a potent activator of NF-κB signalling in NPC[21]. We detected high LMP1 expression in 25.7% of cases by immunohistochemistry and high LMP expression was associated with poor outcome, as previously reported[22]. We identified mutual exclusivity among LMP1 overexpression and NF-κB pathway aberrations dominated by *CYLD, TRAF3, NFKBI and NLRC5* events (Fig. 6; Supplementary Fig. 13). Specifically, genomic alterations in NF-κB pathway regulators were exclusive with high LMP1 expression ($P = 0.00014$; Fisher's exact test, two-tailed), further supporting NF-κB deregulation via somatic genetic events as a core feature of NPC. Our findings provide a potential genetic explanation for the unique inflammatory feature of NPC, contributed by both somatic and viral-mediated addiction to NF-κB signalling in this EBV-associated malignancy[2,5].

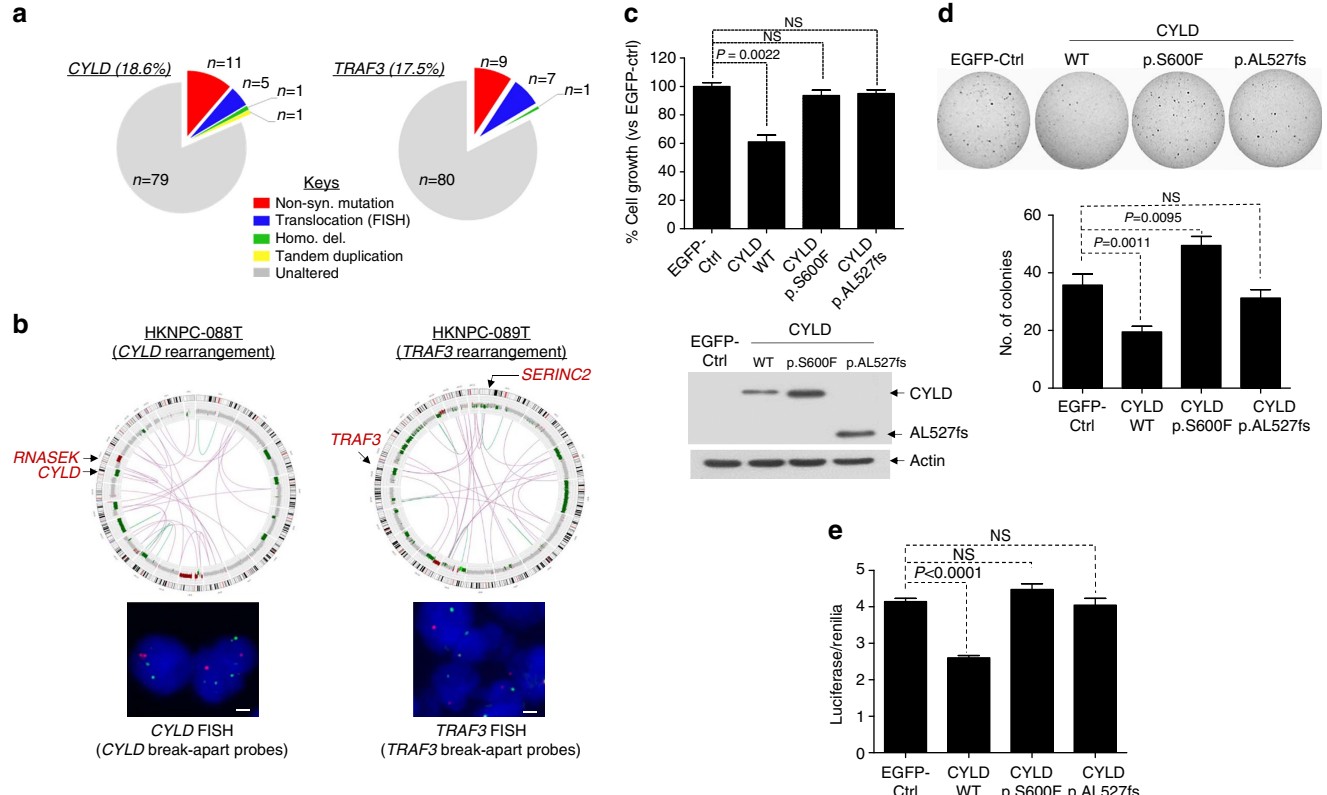

**Figure 5 | CYLD and TRAF3 alterations in NPC. (a)** Additional genetic alterations of *CYLD* and *TRAF3* in NPC detected by fluorescent *in-situ* hybridization (FISH). Left: Five additional cases of *CYLD* rearrangements and 1 case of *CYLD* homozygous deletion were identified by FISH using a *CYLD* break-apart probe and 1 case of *CYLD* tandem repeat was confirmed by Sanger sequencing (Supplementary Figs 7 and 8). Right: Seven additional cases of *TRAF3* gene rearrangements were also identified using *TRAF3* break-apart probes. **(b)** Left: *CYLD* gene rearrangement as revealed by WGS. A circos plot showing the *CYLD* gene rearrangement (HKNPC-088T, confirmed by FISH). A scale bar representing 1 μm is shown in all FISH pictures. Right: *TRAF3* gene rearrangement as revealed by WGS. A circos plot showing the *TRAF3* gene rearrangement (HKNPC-089T, confirmed by FISH). **(c)** Transient transfection of *CYLD* wild-type (WT) gene into an EBV-positive NPC cell line, C666-1 resulted in significant growth inhibition at day 5 ($7 \times 10^4$ cells in 4% FBS, $P = 0.0022$, $n = 6$). Similar results were obtained in three independent experiments. Expression of *CYLD* mutants (p.S600F and frameshift mutant p.AL527fs) revealed a loss-of-function of the tumour suppressive activity versus *CYLD* WT gene in C666-1 cells. CYLD WT and mutant protein expressions are shown underneath. **(d)** *CYLD* WT expression, but not the mutants, inhibited the anchorage-independent growth ability of C666-1 cells in soft-agar colony formation assay. C666-1 cells were infected with retroviral vectors expressing the *CYLD* WT gene and the *CYLD* mutants. Colonies were stained with 0.1% indonitrotetrazolium chloride and counted, and presented as a bar graph, which showed cumulative results of five independent experiments ($n = 15$). **(e)** *CYLD* WT expression was able to reduce NF-κB activity in C666-1 cells in complete culture medium containing 10% FBS. Expression of *CYLD* mutants was not able to suppress NF-κB activity in C666-1 cells. The NF-κB activity was measured using the luciferase/Renilia assay (Promega, USA) with the Cignal NF-κB Pathway Reporter gene system (Qiagen, USA). A cumulative plot of five independent experiments (total $n = 19$) showing changes in NF-κB-dependent luciferase activity in CYLD stable versus EGFP-Ctrl cells.

**Poor outcome for patients with MHC class I aberrations**. In addition to NF-κB pathway alterations, we also noted a high rate of MHC class I gene aberrations (*NLRC5, HLA-A, HLA-B, HLA-C, B2M*). Frequent rearrangements of *HLA-A, HLA-B* and *HLA-C* genes were identified by dRanger analysis[23] (Supplementary Data 7). Twenty-nine NPC cases (30%) were MHC class I-altered (rearrangements and point mutations) (Fig. 7a). Importantly, survival analysis identified inferior outcome in these MHC class I-altered patients with primary NPC (Fig. 7b). It is likely that the deficiency of antigen presentation mechanism may represent an escape mechanism from host immunosurveillance, thus favouring rapid tumour progression. The clinical efficacy of immune checkpoint inhibitors in this genetic subgroup of NPC patients should be examined carefully as these drugs often rely on intact tumour antigen presentation *in situ*.

Given a prior report of an association among ERBB/PI3K activation and inferior outcome in NPC[3], we assessed whether we could confirm this finding in our cohort. We noted no significant difference in outcome among cases with and without known activating mutations in these oncogenic pathways among our primary set of 70 tumours (Supplementary Fig. 14). We did, however, note that mutations in genes known to activate PI3K signalling were non-statistically enriched in recurrent/metastatic cases and found that there was an association between PI3K activating events and poor outcome among recurrent and metastatic NPC patients (Supplementary Fig. 14), in accordance with data presented in HNSCC[24]. This finding could have therapeutic implications in that targeted kinase inhibitors are often explored initially in the recurrent/metastatic setting.

**Somatically altered pathways of NPC**. To better define somatically altered pathways in NPC, we mapped all somatic variants identified in our mutational and copy number analyses to known functional annotations with selected genes, and summarized as a pathway diagram for NPC (Fig. 8). As noted above, frequent

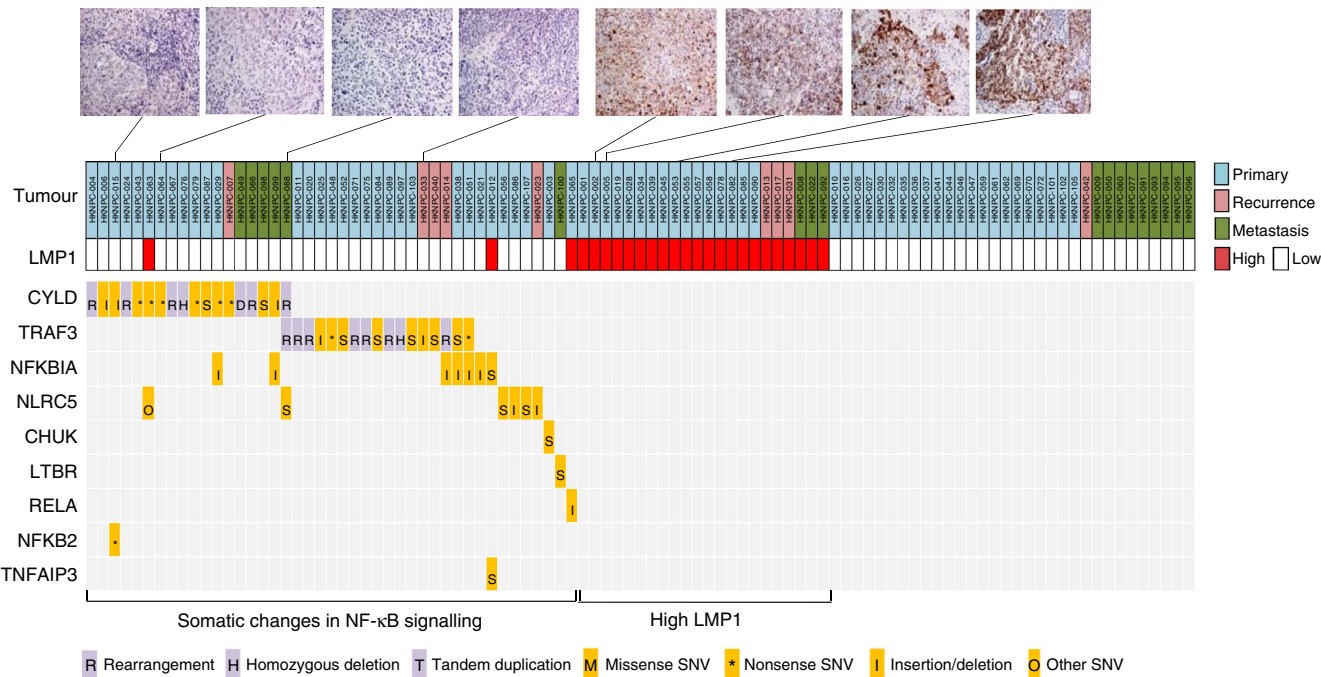

**Figure 6 | Genomic aberrations in NF-κB pathways in NPC.** Mutual exclusivity between LMP1 overexpression and NF-κB somatic alterations in NPC tumours ($P = 0.00014$). LMP1 staining of representative tumours are also shown.

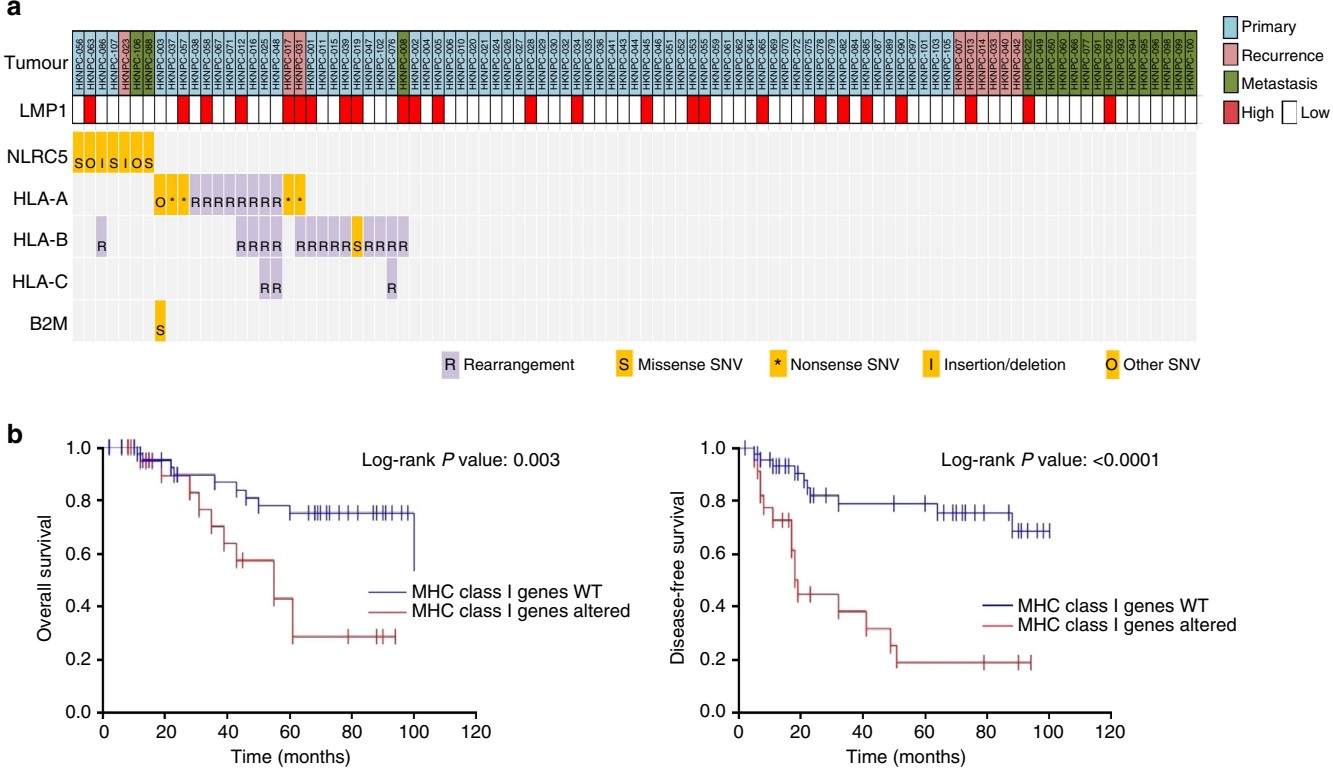

**Figure 7 | Genomic aberrations of MHC class I genes in NPC.** (**a**) A summary of genetic alterations in MHC class I genes, including *HLA-A, HLA-B, B2M* and *NLRC5*. (**b**) NPC patients with MHC class I gene mutations have poorer overall survival and disease-free survival ($P = 0.003$, and $P < 0.0001$, respectively; Log-rank test on 70 primary NPC cases).

somatic alterations in key nodes of the NF-κB pathways were the dominant feature in our NPC cohort, and we also observed recurrent somatic alterations in genes regulating cell cycle progression, chromatin remodelling and DNA repair. Notably, few

variants which have previously been associated with response to a targeted therapeutic agent were identified, although recurrent mutations and amplifications of *RAS* family genes and PI3K/mTOR pathway members were observed.

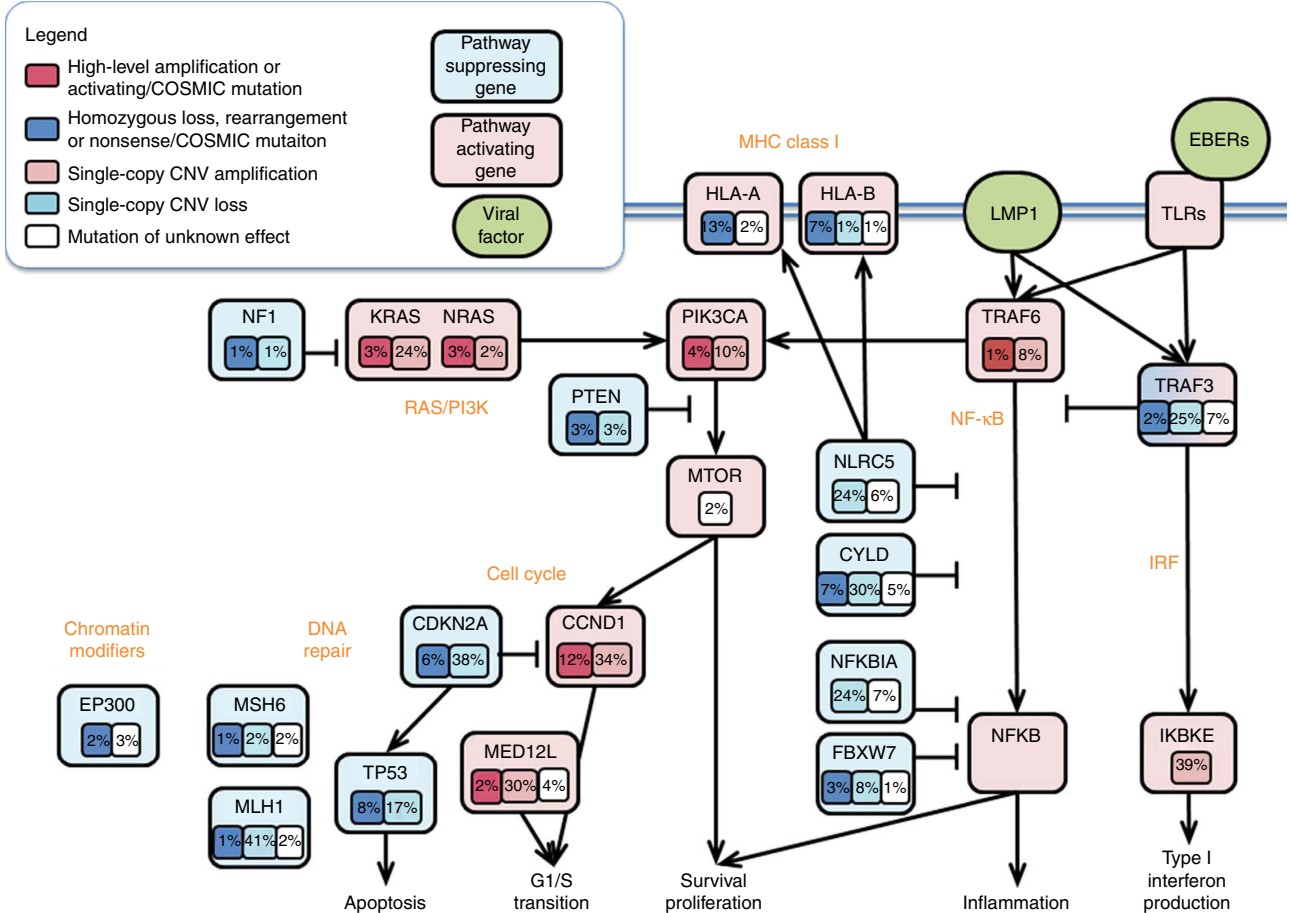

**Figure 8 | Pathway diagram summarizing deregulation of signalling pathways and transcription factors in NPC.** Key affected pathways, components and inferred functions, are summarized in the main text. The frequency (%) of genetic alterations for NPC tumours are shown. Pathway alterations including homozygous deletions, amplifications and somatic mutations are shown. Activated and inactivated pathways/genes, and activating or inhibitory symbols are based on predicted effects of genome alterations and/or pathway functions.

## Discussions

In summary, our study of over 100 micro-dissected NPCs using WES, WGS and FISH revealed a higher degree of somatic mutation and structural variation (SV) in NPC than was previously recognized and enabled the discovery of several novel genomic events that facilitate NPC tumorigenesis, principal among these being NF-κB pathway activation in NPCs with low LMP1 expression. We provide the most comprehensive view of NPC genomics reported to date and identify key somatically altered pathways in this disease including novel mechanisms of NF-κB activation. This study also reveals that NF-κB deregulation, either via somatic genetic events or LMP1 overexpression, is a core feature of NPC. We also demonstrated how NPC genomics can potentially inform therapy, highlighting the importance of targeting activated NF-κB signalling in NPC patients with somatic defects in NF-κB negative regulators. As shown in our previous study, knockdown of NF-κB signals by short interfering RNAs can effectively inhibit NPC growth and survival[14]. It can be envisioned that NF-κB inhibitors and the newly developed small molecule inhibitors targeting Bcl3 can potentially be used as novel therapeutics for NPC patients[25,26]. Furthermore, the discovery of MHC class I lesions in a subgroup of NPC may have biological implications for immune checkpoint inhibitors or other cytotoxic T-cell-based immunotherapies in NPC.

In conclusion, our study suggests that NPCs sub-classified by LMP1 status are distinct in terms of clinical history and outcome as well as genomic features. Future laboratory studies and precision medicine-based treatment approaches for NPC would need to consider the features of these distinct subclasses.

## Methods

**NPC tissue collection.** All NPC tumours were collected by endoscopy or surgery. Patient consent was obtained according to institutional clinical research approval (IRB) at the Chinese University of Hong Kong, Hong Kong (for Asian cases), and at the University of Pittsburgh, USA (for US cases). The original clinical and survival data of patients were provided in Supplementary Data set.

**Exome sequencing and variant calling.** DNA was extracted from microdissected FFPE or frozen tissue samples using the Qiagen FFPE DNA extraction kit as per the manufacturer's protocol. Solution based hybrid capture with the Agilent Sure-Select method was followed by sequencing on the Illumina Hi-Seq instrument to a goal coverage of 80 × as previously described[4]. Raw FASTQ files were processed for downstream analysis using PicardTools (http://broadinstitute.github.io/picard/). BAM files are deposited to dbGAP-NHGRI (Study ID: 20055, Nasopharynx Cancer Whole-Exome Sequencing) for public use.

Variant calling (SNVs, indels) was performed using the Firehose pipeline running Mutect[27], then filtering out OxoG artifacts and likely germline mutations that were previously seen in both dbSNP build 134 and 1,000 Genome data. To remove potential artifacts from FFPE samples, we filtered out variants in a variant frequency peak with <5% frequency and total coverage under 30 × . The overall mutation rate and average coverage were calculated for the 105 tumour and normal pairs. Significance analysis was conducted using a local version of MutsigCV 2.0, which considers gene expression, replication time and chromatin state when calculating the background rate. To further improve our confidence in the most significantly mutated genes, we also called mutations using Strelka[28] and filtered out genes for which <25% of mutations overlapped between the two methods. To assess for the most likely cancer-associated mutations, we chose mutations seen in at least three samples in the COSMIC database using Oncotator (10.1002/

humu.22771). Germline variants identified from NPC patients were shown in Supplementary Data 8. We used the Fluidigm Access Array microfluidic system, which allows for multiplexed, high-throughput, amplicon sequencing using custom targeted primers for a total of 464 positions to validate a total of 82% of variants (Supplementary Data 9). These variants included both SNVs and indels in the cases with minimum original allele frequencies of 15%. Copy number analysis was performed using the ReCapseg, which calls and segments targeted genome data, using panel of normals to model noise and normalize calls. Oncotator was used to annotate genes affected by the CNVs. Significantly recurring CNVs were determined by putting the CNV segments into GISTIC 2.0 with mostly standard settings, except for 0.6 as the amplification and deletion threshold due to our samples being microdissected, and join segment size of 10 and $q$ value threshold of 0.01 to reduce noise. Somatic rearrangements were detected using dRanger[23], using a score cutoff of 3. Mutational signature analysis was conducted using the SomaticSignatures package in R (10.1093/bioinformatics/btv408) to identify the mutation spectrum, and assess the number of signatures using non-negative matrix factorization method. From simulating 2 to 10 signatures, we found the best-fit model consisted of four signatures, with the lowest standard error and describing 77% of variance. These signatures were correlated to known mutational processes from the COSMIC database (version 8.12.2015.txt) using Pearson's correlation coefficient. As we increased our model to more than four signatures, we did not identify any additional highly correlated COSMIC signatures.

**Whole-genome sequencing.** For the 15 NPC cases that were subjected for WGS, 1 µg of genomic DNA extracted from the microdissected frozen tumours and matched normal samples were subjected to the Illumina Whole-Genome Sequencing Service in Macrogen (Seoul, Korea). Standard Illumina protocols and Illumina paired-end adapters were used for library preparation from the fragmented genomic DNA. Sequencing libraries were constructed with 500 bp insert length. WGS was performed using the Illumina HiSeq 2000 platform with a standard 100-bp paired-end read previously described[29]. Mean target coverage of 40 × and 60 × was achieved for the normal and tumour samples, respectively. The raw sequence reads were processed and aligned to the hg19 human reference genome using Isaac aligner[30]. BAM files are deposited European Nucleotide Archive (ENA) (Accession number PRJEB12830, WGS of matched normal and tumour samples of NPC patients). Identification of somatic SNV and SV was conducted by MuTect[27] and Manta[31], respectively. To evaluate the sensitivity of somatic SV results, we used only one SV result when different calls share breakpoints within 1,000 base pairs. We predicted somatic copy number aberration and loss of heterozygosity in NPC genome using TITAN[32]. Germline heterozygous SNVs of normal sample and the tumour read counts of those sites were extracted by samtools/bcftools[33]. The identified somatic copy number aberrations and SVs of each NPC were visualized by CIRCOS[34].

**Pathway diagram for NPC.** To calculate recurrent percentages, we considered an alteration to be activating if it landed in a known oncogene and was either a COSMIC mutation (in at least three samples) or highly amplified. Similarly, we considered alterations to be inactivating if they were nonsense or frameshift or COSMIC mutations or homozygous deletions. Highly recurrent single copy gains or losses were considered potentially activating or inactivating, respectively. Missense or other nonsynonymous mutations were taken into consideration as unknown functional effect.

**Fluorescence *in situ* hybridization.** FISH analysis was conducted to detect the *CYLD*, *TRAF3* and *NFKBIA* in FFPE NPC tissue sections. Commercially available dual-colour (orange and green) flanking probes of *CYLD*, *TRAF3* and *NFKBIA* were used (Empire Genomics, USA). FISH analysis was performed as we previously described[35]. In each case, at least 100 nuclei were analysed. The break-apart signals were scored if orange and green signals separated by one signal diameter or only single signal were detected. Cases with translocation of *CYTD*, *TRAF3* and *NFKBIA* were defined as >20% of breakapart signals in the tumour cells.

**LMP1 immunohistochemical staining.** The expression of LMP1 was determined in FFPE NPC sections by immunohistochemical staining. After de-waxing, the sections were subjected to antigen retrieval and staining in the automated slide processing system BenchMark XT (Ventana Medical systems Inc., Tucson, AZ) with the OptiView Amplification kit (Ventana Medical Systems Inc.). The primary antibody used in this study was anti-LMP1 mouse monoclonal antibody (CS.1-4, Dako). The LMP1 expression was assessed by assigning a proportion score and an intensity score[36]. The proportion score was according to the percentage of tumour cells with positive membrane and cytoplasmic staining (0–100). The intensity score was assigned for the average intensity of positive tumour cells (0, none; 1, weak; 2, intermediate; 3, strong). The LMP1 staining score was the product of proportion and intensity scores, ranging from 0 to 300. The LMP1 expression was categorized into absence/low (score 0–100) and high (score 101–300).

**Immunoblotting.** Immunoblotting was performed as previously published[37]. All antibodies used were from Cell Signaling Technologies, USA. These are antibodies against CYLD (#12797) and actin (#8456).

**Generation of CYLD mutant cells.** The *CYLD* wild-type (WT), and various *CYLD* mutants were generated by gene synthesis and cloned into the pMXs-puro vector (Cell Biolabs, USA). The pMXs-puro-EGFP plasmid was used as control. The retroviral packaging cell line, Plat A cells were used to generate infective retroviruses for C666-1 and HK1-EBV infection according to the manufacturer's instructions. Seven days after infection, cells were collected for the confirmation of CYLD mutant expression by immunoblotting. For growth assay, cells were plated at low density ($1 \times 10^4$ per well) in RPMI (for C666-1) or DMEM (for HK1-EBV) with 4% FBS and assessed for proliferation at 48 h. Mutant-induced cell growth, compared with the *EGFP* vector control or *CYLD* WT, was analysed by MTT assay. C666-1 and HK1-EBV cell lines were previously established and authenticated in our laboratory. No mycoplasma contamination was detected in these cell lines. An uncropped images of western blot membranes for the expression of CYLD WT and mutant protein is shown in Supplementary Fig. 15.

**Soft agar colony formation assay.** C666-1 stable cells expressing the CYLD WT and various *CYLD* mutants were plated at a density of $1 \times 10^5$ cells per well of a 6-well plate. Cells were grown in 0.35% soft agar for 21–24 days. Colonies were then stained with 0.1% indonitrotetrazolium chloride (Sigma-Aldrich, USA) for 48 h, and counted under a stereomicroscope.

**Luciferase assay.** C666-1 cells were co-transfected with CYLD WT or various CYLD mutants with Cignal NF-κB Pathway Reporter Assay (Qiagen, USA). After 48 h, luciferase activity was measured using the Dual-Luciferase Reporter Assay (Promega, USA). Luciferase counts were normalized to Renilia counts for each sample.

**Immunofluorescence staining for Bcl-3 translocation.** C666-1 cells were transient transfected with the plasmid constructs expressing either the HA-tagged CYLD WT or various *CYLD* mutants. At 48 h post-transfection, cells were fixed with 4% paraformaldehyde in PBS for 10 min. Cells were then permeabilized and blocked with blocking buffer (3% BSA, 0.1% Triton X-100 in PBS). After washing with PBS, cells were incubated with the corresponding primary antibodies (Anti-HA.11, 1:1,000 from BioLegend, USA; Anti-Bcl3 (C-14), 1:400 from Santa Cruz, USA), and then in fluorophore-conjugated secondary antibodies (Alexa Fluor 488 and Alexa Fluor 555 from Invitrogen, USA). Cells were finally washed with PBS and counterstained with 4,6-diamidino-2-phenylindole to label the nuclei. The fluorescence images of cells were captured with the Axio Observer microscope (Zeiss) equipped with a cooled CCD (charge-coupled device) camera.

**In vivo ubiquitination assay.** HeLa cells were transfected with various CYLD-expressing plasmids as indicated. Cells were harvested at 24 h post-transfection and total protein extract was extracted with RIPA buffer supplemented with protease inhibitors. Cell lysate was then boiled for 30 min in RIPA buffer containing 1% SDS to disrupt protein-protein interaction. The cell lysate was then cooled down on ice and further diluted with RIPA buffer to decrease the SDS concentration to 0.1% before performing immunoprecipitation. One milligram of protein lysate was incubated with 1 µg anti-FLAG (M2) antibody from Sigma-Aldrich overnight at 4 °C with constant rocking. Protein G-conjugated Dynabeads (Novex, Invitrogen) was then added to pull-down the protein-containing antibody complex. The beads were washed with RIPA buffer three times and incubated in 2 × Laemmli buffer at 70 °C for 10 min to dissociate the Protein–antibody complex. The protein samples were then separated by SDS–polyacrylamide gel electrophoresis and western blotting was performed using the antibodies as indicated. The His-tagged K48R ubiquitin plasmid was a kind gift from Dr Ramin Massoumi (Lund University, Sweden).

**Statistical analysis.** Statistical analyses to test for correlation between genomic and clinical features such as age, sex and clinical staging were performed using standard R packages. We used the Fisher's exact test for discrete variables, the Log-rank test for continuous variables, and the Bonferonni method of multiple testing correction.

**Data availability.** The WES and WGS data that support the findings of this study have been deposited in dbGAP-NHGRI (Study ID: 20055, Nasopharynx Cancer Whole-Exome Sequencing) and European Nucleotide Archive (ENA) (Accession number PRJEB12830), respectively. All remaining data are included in the manuscript and Supplementary Informations or available from the authors upon request.

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

## Acknowledgements

This study is supported by Core Utilities of Cancer Genomics and Pathobiology, Theme-based Research Scheme (T12-401/13-R), Research Grant Council, Hong Kong. K.-W.L. is funded by Research Grant Council, Hong Kong (#471413, 470312, 471211, 1404415, 14138016, General Research Fund), Focused Innovations Scheme and Faculty Strategic Research (4620513) of Faculty of Medicine, and VC's One-off Discretionary Fund (VCF2014017, VCF2014015), The Chinese University of Hong Kong. S.-W.T. is supported by AoE/M-06/08. V.W.Y.L. is funded by the Research Grant Council, Hong Kong (#17114814, General Research Fund), and the Start-up Fund from the School of Biomedical Sciences, Faculty of Medicine, the Chinese University of Hong Kong. K.-F.T. is funded by the Research Grant Council, Hong Kong (General Research Fund #14138016 and #14104415). M.L.H. is supported by NCI 1F30CA180235. P.S.H. is supported by NCI K08 CA163677.

## Author contributions

Y.Y.L., G.T.Y.C., V.W.Y.L., P.S.H., K.-W.L. designed the study and interpreted the data. Y.Y.L., V.W.Y.L., K.Y.Y., J.S., M.R., N.G.C., S.-K.Y., J.-Y.S. S.-D.L., P.S.H., K.-W.L. contributed to data analyses of cancer genomes. K.-F.T., B.B.Y.M., J.K.S.W., E.P.H., A.W.H.C., J.Y.K.C., P.B.S.L., L.W., A.M.V.C., S.I.C., M.L.H., S.-W.T., A.C.v.H., A.T.C.C., J.R.G., K.-W.L. provided study materials, recruited subjects and participated in diagnostic evaluations. Y.Y.L., G.T.Y.C., V.W.Y.L., C.C., M.K.F.M., Y.Y.Y.O., M.H.N.L., P.P.Y.L., Z.W.Y.L., H.-L.N., P.-M.H., K.R.V., P.H.Y.P., S.W.M.L., L.J., C.-Y.F., S.-T.H. performed the sequencing and experiments. Y.Y.L., V.W.Y.L., P.S.H., K.-W.L. wrote the manuscript.

## Additional information

**Competing financial interests:** V.W.Y. serves as a scientific consultant for Novartis, Hong Kong. B.B.Y.M. received research grant and serves the advisory board from Novartis, Hong Kong.

**Publisher's note**: 

