## [Peer Review File · Nature Communications]

Reviewers' comments:

Reviewer #1 (Remarks to the Author): Expert in genomics

In this manuscript, Li et al. performed WES on 111 micro-dissected nasopharyngeal tumors, including primary, local recurrent and metastatic tumors. They also sequenced the whole genomes of 15 such tumors. With micro-dissection, the authors showed that they could increase the power to discover somatic mutations in these biopsies. However, the authors did not mention whether they have performed orthogonal validations, which clearly should be done to estimate the accuracy of calling the somatic mutation. In particular, both FFPE and frozen tissue samples were used in this study. However, no detailed information was provided regarding which samples were from which group. The mutational rate of FFPE samples is known to tend to be higher and likely produce more false positive calls comparing with frozen tissues. This needs to be addressed comprehensively in order to determine the true mutational load of nasopharyngeal tumors. In addition, enrichment for genomic aberrations of multiple negative regulators of the NF- κ B pathway, including CYLD, TRAF3 and NFKBIA were observed. Functional evidence of the tumor suppressing role of CYLD was provided. While the results are clear and provide novel observations, I have several additional comments (listed below) that need to be addressed.

1) In Supplementary Fig. S3, the authors noted that many mutations with previously annotated roles in cancer were not shared between paired primary and recurrent lesions. This disagrees with a number of published studies showing that the majority of driver mutations are likely to be shared between primary and recurrent tumors. The lack of shared mutations in this figure might reflect the relatively low sequencing depth (86X). A more precise conclusion can be achieved by targeted resequencing of the identified mutations at much higher depth.

2) It is known that the copy number estimations from WES data are less robust than WGS. FFPE samples might further decrease the accuracy of the estimation. For example, Figure 2C (from WES) shows many more copy number changes than figure S5. Therefore, more rigorous validations of copy number changes from WES data need to be performed, especially those involving NF- κ B factors highlighted in the manuscript (Figure 2a).

3) In FISH experiments, normal controls without rearrangements or deletions should be included. Also, scale bars are needed.

4) In the functional analysis of the CYLD gene, the C666-1 cell line is an ideal model as it contains a disrupted allele. It will be interesting to examine if restoration of CYLD in this cell line can suppress cell malignancy. Also, loss-of-function experiments should be performed in NPC cell lines to confirm further its anti-tumor effects.

5) In Figure 3c: the Y Axis was labelled differently, but based on the description of methods, they all appeared to be "MTT assay". If so, then the middle panel is repeating the upper panel, and the label needs to be unified. In addition, soft-agar colony formation assay may generate more conspicuous changes than MTT assay. "*" needs to be defined.

6) The changes in the first 4 groups of Figure 3d are very trivial, albeit with a significant p value. With such small changes, I am not sure what the biological significance is? Maybe loss-of-function experiments can show a clearer picture?

7) The data suggests that CYLD can only suppress NF- κ B activity in the absence but not the presence of TNF- α treatment (Figure 3d)? What might be the reason?

8) CYLD, S600F and AL527fs each seem to have different roles in the absence of TNF- α treatment, but the same role in the presence of TNF- α treatment (Figure 3d)?

9) Survival data of the patients should be provided in the manuscript.

Reviewer #2 (Remarks to the Author): Expert in Nf κ B pathway

Li and colleagues performed whole exome sequencing on 111 tumor/normal nasopharyngeal carcinomas, a rare head and neck cancer. Given the small amount of available genomic data previously reported for this particular cancer along with the fact that this cohort has clinical data, makes this a valuable genomic data set for the broader research community.

- 1) I don't see a table listing all the somatic mutations. This will be essential for the community to have this information reported in a supplemental table. It should also include the gene, protein altering event if any, protein mutations caused, variant allele frequency, whether its been reported in COSMIC etc.
- 2) Any rare germline mutations for the genes highlighted in the paper should also be listed in a supplemental table.
- 3) It was interesting that mutation burden was prognostic of survival but there was no correlation with other clinical factors. A Cox proportional hazard model should also be built using the mutational burden data along with the other clinical factors to evaluate the prognostic value of these factors.
- 4) The methods section for mutational signatures needs to be expanded. The SomaticSignatures package was used but the details of what was done are not given. Was NMF used to derive 30 signatures or was another number used? Was the number of signatures to use assessed? Which data (beyond this data set) was included to derive the signatures? How were the underlying causes (e.g. mismatch repair) of these signature applied, manual observation and comparison?
- 5) Was TP53 mutation status (or any of the other significantly mutated genes) prognostic of survival?
- 6) As the majority of CYLD mutations are in the DUB domain, the authors should check the expression status and stability of CYLD substrates (RIP1, TRAF2, Bcl3) on endogenous level.
- 7) A lot of mutations in TRAF3 are cysteine mutants, which could affect TRAF3 protein expression and stability. The authors should also check p100 (alternative NF- κ B) processing, which is affected by TRAF3 expression.
- 8) Data in figure 3d show minimal effect making it hard to correlate this effect to cell growth effect. Is cell viability affected?
- 9) The authors should examine LMP1 expression by western blotting to support the data in figure 4a, especially for WT protein in high expression cases.
- 10) Please increase the font size for text in the figures - it is barely legible.

Reviewer #3 (Remarks to the Author): Expert in HNSCC

The manuscript details sequencing analyses of 111 micro-dissected EBV-positive nasopharyngeal carcinoma samples (predominantly Asians). In 41% of cases the authors discovered enrichment for genomic aberrations in negative regulators of the NF- κ B pathway, such as CYLD, TRAF3, NFKBIA and NLRC5. Functional analysis confirmed inactivating CYLD mutations as drivers for NPC cell growth. Furthermore, the authors found that the expression of latent member protein 1 (LMP1), an EBV oncoprotein that constitutively activates NF- κ B signaling, was mutually exclusive of NF- κ B pathway aberrations and LMP1-overexpression, suggesting that NF- κ B activation is selected for by both somatic and viral events during NPC pathogenesis. The data presented in this article also suggests that NPC tumors subclassified by LMP1 status are distinct in terms of clinical history and outcome and genomic features. Overall, the study gives a comprehensive view of NPC genomics in an Asian population and identifies key altered pathways in this disease. It is, however, somewhat hampered by presentation of the 'pathway' analyses and functional experiments. There are several points which would improve the manuscript in terms of clarity and impact.

- * All signatures derived from the non-negative matrix factorization (NMF) analysis should be shown in Supp. Figures.
- * Figure 2a - please explain what the asterisks stand for.
- * In 2 tumors the authors detected 2051 and 817 somatic coding mutations, and link the reader to

Figure 2a. However, the axis Y extends only up to around 500 mutations per sample. The authors have to re-plot the figure or explain this in the legends.

* Supp. Figure 3 - please explain what the arrows emphasize. The arrows are not mentioned neither in the text nor in the legends.

* In addition to the data shown in Supp. Figure 3, please add a Supplemental Table listing all the mutations shared between the paired primary/recurrent and primary/metastatic NPC. Also, were silent/synonymous mutations considered?

* Page 10 - "one case harboring homozygous deletion confirmed by Fluorescence in situ Hybridization (FISH; Supplementary Fig. S7)". It appears the wrong figure is referenced. FISH is actually shown in Supplementary Figures S8 and S9, not S7. Figure S7 as it appears in the supplemental material is not mentioned in the text. Should be corrected.

* It seems that HK1-EBV cells are already expressing relatively high levels of CYLD, therefore, it is not a good cell line model for overexpression experiments. The functional experiments shown in Figure 3C should be repeated in a cell line depleted for CYLD or at least expressing a low CYLD level. For example, the biological role of wt-CYLD or S600F CYLD can be assessed in C666-1 cells, which do not express CYLD protein. It would also be interesting to see if CYLD knockdown in HK1-EBV cells will result in the same phenotypical changes as those seen in cells expressing the mutant variants.

* Mapping of the altered genes to pathways is an important characteristic of the study but the main text provides little information. It is not clear what software or model the authors used for pathways analysis. The methods section entitled "Pathway analysis" lacks any explanation about the pathways analysis and has to be significantly expanded. There is little statistical support given for any sort of enrichment analyses described in the text.

* Was there any orthogonal validation done on any of the variants detected?

* For the variants called, the authors should provide read counts for non-synonymous mutant and reference for each relevant lesion at that position (as a Supplemental excel Table). As it stands, there is only information given about average coverage of 86X. This will be useful in understanding the quality of the data being presented.

* Information sharing strategies are not discussed in this paper and should be added.

* Mutational nomenclature should be in standard format, that is c. and p.

* Other than EBV status, there are no patient characteristics described that could also relate to the disease and the mutational burden. For example, were these smokers? Alcohol intake? If this information is available it should be added.

* The manuscript should be formatted in accordance to the journal's requirements. The abstract is missing and should be added. The titles for new paragraph sections should be added. A formal discussion section should be added.

Point-by-Point Reply:

We thank the reviewers' for their positive comments on the study, stating that "the results are clear and provide novel observations" (Reviewer #1), "this particular cancer along with the fact that this cohort has clinical data, makes this a valuable genomic data set for the broader research community" (Reviewer #2), and "the study gives a comprehensive view of NPC genomics in an Asian population and identifies key altered pathways in this disease" (Reviewer #3). We are grateful to the reviewers' expert comments to further improve the manuscript in terms of clarity and impact. We have, therefore, revised the manuscript addressing all the comments of the reviewers (in red highlights in the text), as well as listed as "point-by-point reply" below for your kind consideration.

Reviewer #1:

Overall comments:However, the authors did not mention whether they have performed orthogonal validations, which clearly should be done to estimate the accuracy of calling the somatic mutation. In particular, both FFPE and frozen tissue samples were used in this study. However, no detailed information was provided regarding which samples were from which group. The mutational rate of FFPE samples is known to tend to be higher and likely produce more false positive calls comparing with frozen tissues. This needs to be addressed comprehensively in order to determine the true mutational load of nasopharyngeal tumors.

Our Reply: Thank you for the reviewer's expert suggestion and comment. To call somatic mutations we used MuTect, an established algorithm developed at the Broad Institute of Harvard and MIT optimized for mutation calling in stromally contaminated tumor tissues, including FFPE samples, with high validation rates reported in various studies to date including the original report of Cibulskis *et al.* (Nat Biotechnol. 2013 Mar;31(3):213-9) as well as Van Allen *et al* (Nat Medicine, 2014 Jun; 20(6): 682–688) which specifically addressed the use of this method in FFPE samples. While we are not aware of a consensus method for orthogonal validation of WES mutation calls from stromally admixed FFPE samples given the challenge of matching the sensitivity of hybrid capture, especially for mutation calls at lower allele fractions, we have now added additional validation of mutation calls using the Fluidigm method.

Specifically, we used the Fluidigm Access Array microfluidic system, which allows for multiplexed, high-throughput, amplicon sequencing using custom targeted primers. We focused on potentially cancer-related somatic mutations that were either discussed in the manuscript or seen in at least three samples in the COSMIC database, and all called somatic variants in trios. We designed primers for a total of 464 positions on 2 Fluidigm

chips. Raw FASTQ result files were processed and aligned using bowtie2 on standard settings. Overall, we had an average of 216,649 +/- 48,880 reads, 85% +/- 7% alignment rate and coverage of 660X +/- 181 across the 74 samples.

We manually reviewed variants using IGV and found that we validated 82% (242/294) of variants, including both SNVs and indels in the cases with minimum original allele frequencies of 15% and 71% of mutations (264/373) at a 5% allele fraction cut point (table below). We felt it was reasonable to see that the validation rate increased, and the number of uncovered positions decreased as the original allele frequency of detected variants increased. More specifically for the trios (matched normal also sequenced), we were able to validate the presence of similar shared variants from WES in HKNPC012, HKNPC008 and HKNPC009. These results recapitulate what we observed in our manuscript.

minimum original AF	# validated in tumor validation sample (and not in normal, where available)	#not validated (not present in validation tumor sample)	not covered in one of the validation samples	validation %
15%	242	52	50	82%
10%	298	85	67	78%
5%	264	109	77	71%

Per reviewer's suggestion, we have also updated our **Supplementary Data** (Clinicopathological details of NPC patients (N=105 individuals)) to indicate which of the samples were originally from FFPE or fresh frozen microdissected NPC specimens.

Comment 1: In Supplementary Fig. S3, the authors noted that many mutations with previously annotated roles in cancer were not shared between paired primary and recurrent lesions. This disagrees with a number of published studies showing that the majority of driver mutations are likely to be shared between primary and recurrent tumors. The lack of shared mutations in this figure might reflect the relatively low sequencing depth (86X). A more precise conclusion can be achieved by targeted resequencing of the identified mutations at much higher depth.

Our Reply: Thanks for the reviewer's comment. We acknowledge the limitation of the very small number of microdissectable paired primary/recurrent tumors in our current report (only 2 cases available: HKNPC001 and HKNPC012) since WES of primary NPC tumors was our focus in this study and that a much larger cohort would be required to address this issue

satisfactorily. We would note that a recent study of head and neck squamous cell carcinomas which we contributed to reported a lower rate of shared mutations among metachronous tumors as compared to synchronous tumors (Hedberg et al., J Clin Invest. 2016 Jan;126 (1):169-80) as is discussed in detail below.

In brief, our WES showed that HKNPC001 carried a total of 7 shared mutations between the primary and recurrent tumors. Both primary and recurrent tumors harbored the hotspot NRAS (Q61R) mutation. For HKNPC012, 16 shared mutations [including SYK(A412T)] were identified between the primary/recurrent pair, while an HRAS(Q61R) hotspot mutation was only identified at recurrence.

Using the cancer genes defined by the Cancer Gene Census (COSMIC, UK), only a small number of these (7 and 16 shared genes from the HKNPC001 and HKNPC012 primary/recurrent pairs) were previously annotated to have known roles in cancer. Interestingly, our findings in recurrent NPC seem to concur with recent data reported in primary/recurrent HNSCC pairs that only TP53 mutations were the only shared annotated cancer gene (Hedberg et al, J. Clin. Invest., 2016 Apr 1;126(4):1606) between the each of the primary/recurrent HNSCC pairs (N=9 cases in total). All other identified shared primary/recurrent HNSCC mutational events were in *DDR2*, *ASPH*, *ATP1A2*, *PRLR* (only 1 case each), *MUC16* and *SYNE1* (2 cases each), which have not been classified as cancer genes (COSMIC). This may reflect our current insufficient or incomplete annotation of all potential cancer genes in human carcinogenesis and/or insufficient sequencing coverage as the reviewer notes. Another possibility is that our concurrent findings in both recurrent HNSCC and NPC may suggest the emergence of some new and rather genetically distinct subclones during the recurrences of these aggressive head and neck cancers. In fact, a recurrent HNSCC carried a newly emerged TP53 mutation was noted in one case, while another recurrent tumor carried a newly emerged DDR2 mutation was noted in HNSCC (Hedberg et al, J. Clin. Invest., 2016 Apr 1;126(4):1606), a pattern that is consistent with our NPC case HKNPC012. We have therefore modified our text accordingly to indicate these possibilities (p.9-10 of the revised manuscript).

For easy reference, we have also listed all the mutational events (including silent and non-silent mutations with allele frequencies) for the paired NPCs: primary/recurrent, primary/metastatic, and metastatic site 1/metastatic site 2 in the supplementary data ("Mutational events of paired NPCs: primary/recurrent, primary/lymph node metastasis, and metastatic site 1/metastatic site 2" file). Furthermore, by multiplexed amplicon sequencing, we also assessed mutations in the primary or recurrence/metastasis only, and shared mutations in HKNPC008 (met site 1/site 2), HKNPC009 (met site 1/site 2) and HKNPC012 (primary/recurrence) in our validation experiments (HKNPC001 did not have sufficient material remaining for additional DNA extraction).

Trio HKNPC012 (primary/recurrence) displayed an overall validation rate of 80% when

comparing WES and Fluidigm calls at 160 positions. Of the subset of mutations assessed in both sequencing experiments 5% were shared among both tumors in Fluidigm and 8% in WES. These data agree with the lack of concordance among the primary and recurrent lesion in this single case.

Trio HKNPC008 (met site 1/ site 2) mutations validated at 79% at 72 covered sites with a shared mutation rate of 56% at the validated sites in Fluidigm and 74% in WES. Interestingly, sites not chosen for validation had a lower shared rate in the WES which might be expected as COSMIC mutations were selected for validation. Trio HKNPC009 (also met site 1/ site 2) had the fewest number of sites available and covered for validation (33) with a validation rate of 67%. Of these 68% were shared in Fluidigm and 86% in WES.

Future studies on a larger panel of paired primary and recurrent tumors will allow us to have a clearer picture of the genetic and clonal evolution of NPC.

Comment 2: It is known that the copy number estimations from WES data are less robust than WGS. FFPE samples might further decrease the accuracy of the estimation. For example, Figure 2C (from WES) shows many more copy number changes than figure S5. Therefore, more rigorous validations of copy number changes from WES data need to be performed, especially those involving NF-KB factors highlighted in the manuscript (Figure 2a).

Our Reply: According to the reviewer's suggestion, we have validated 44 copy number variants identified from our WES data by FISH analysis using gene specific probes. These CNVs included the amplification of *CCND1* and homozygous deletion of *CDKN2A* which have been reported to be critical genetic changes in NPC tumorigenesis. The copy number changes of 4 NF-kappaB negative regulators, *TRAF3*, *NFKBIA*, *CYLD* and *NLRC5* were validated in the samples in which FFPE sections were available. As shown in Supplementary Figure S6 almost all selected CNAs identified by WES (43/44, 97.7%) were confirmed by FISH analysis with gene specific probes. In HKNPC-076, WES data only detected copy number loss of *CYLD* while homozygous deletion was shown in FISH analysis. Furthermore, while we compared the CNVs identified in the 15 frozen NPC samples with both WES and WGS data, the patterns were highly consistent. The findings supported the accuracy of our CNVs calling in the current WES study.

Comment 3: In FISH experiments, normal controls without rearrangements or deletions should be included. Also, scale bars are needed.

Our Reply: Normal controls without rearrangements and scale bars have been added onto all FISH data in this revised manuscript (Fig 5b, Supplementary Figures S6, S7, S8, and S12). The respective figure legends have been modified.

Comment 4: In the functional analysis of the *CYLD* gene, the C666-1 cell line is an ideal model as it contains a disrupted allele. It will be interesting to examine if restoration of *CYLD* in this cell line can suppress cell malignancy. Also, loss-of-function experiments should be performed in NPC cell lines to confirm further its anti-tumor effects.

Our Reply: We thank the reviewer's expert comment. In our original manuscript, we used HK1-EBV cells for the functional study of *CYLD*. We fully agree with the reviewer that the C666-1 cell line contains a disrupted allele of *CYLD* and thus does not express any detectable endogenous *CYLD* protein. With C666-1's clean background with no *CYLD* protein expression, we repeated the functional growth experiment on *CYLD* wildtype and *CYLD* mutants (S600F, and AL527fs) as we did previously with HK1-EBV cells. As shown in this new Figure 5c, *CYLD* WT did suppress C666-1 cell growth ($P=0.0022$) and expression of mutants confirmed that they were loss-of-function mutants and could not suppress NPC cell growth. These results were consistent our findings in HK1-EBV (please note that the original results of *CYLD* expression on HK1-EBV cell growth are now moved to new Supplementary Figure S11).

We have performed *CYLD* siRNA experiments on both HK1-EBV and C666-1 cells. However, since C666-1 cells do not express any endogenous *CYLD*, there was no observable changes upon *CYLD* siRNA transfection vs. control siRNA (data not shown). For HK1-EBV, we also observed no significant change in cell growth over 72 hours (data not shown). This is likely due to the lack of p50/p50/Bcl3 nuclear NF- κ B signals in this EBV-reinfected NPC cell line. Furthermore, as shown in Supplementary Figure S12, *CYLD* WT expression in C666-1 cells can inhibit Bcl3 ubiquitination and nuclear accumulation, which is known to be important for its growth-promoting effects. The respective text has been updated (p.12 of this revised manuscript).

Our results indicate that *CYLD* mutants are functionally distinct from *CYLD* knockdown, suggesting the *CYLD* mutants are likely to be exerting some functional effects or dominant negative effects (not shared by simple knockdown) which contribute to C666-1 cell proliferation and anchorage-independent growth phenotypes observed (demonstrated in C666-1 cells as suggested by the reviewer). Dominant negative effects of *CYLD* mutant have been suggested previously (Miliani de Marval P, *et al*, Cancer Prev Res (Phila) 2011 Jun; 4(6): 851-9).

Comment 5: In Figure 3c: the Y Axis was labeled differently, but based on the description of methods, they all appeared to be "MTT assay". If so, then the middle panel is repeating the upper panel, and the label needs to be unified. In addition, soft-agar colony formation assay may generate more conspicuous changes than MTT assay. "*" needs to be defined.

Our Reply: We appreciate the reviewer's suggestion to improve data presentation. Per reviewer's suggestion above, the original Figure 3c has been replaced with our new C666-1 data, in which all superscripts "*" for statistical significance have been replaced by the calculated P-values to avoid any confusions (new Figure 5c, as well as Figure 5d & 5e). Note that in our new data using C666-1 cells as suggested by the reviewer, we observed a significant change in cell growth ($P=0.0022$). Please note that our original Figure 3 becomes our new Figure 5 due to the request of another reviewer for enlargement of figures for figure clarity purposes only. Further, all Y-axes have been re-labeled in a unified way (new Figure 5c for C666-1 growth, and Supplementary Figure S11 for original HK1-EBV growth). All experiments were performed with reproducible results in 3 independent experiments.

Soft agar colony formation assays were performed using stable C666-1 stable cells expressing the control EGFP, CYLD WT and CYLD mutants (S600F and frameshift mutant AL527fs). Our data showed that CYLD WT suppressed anchorage-independent growth of C666-1 cells for 45.4% when compared with the EGFP control, while the mutants were not capable of suppressing C666-1's anchorage independent growth ability, confirming these mutants to be loss-of-function mutants.

The respective text has been updated (p.11-12 of this revised manuscript).

Comment 6: The changes in the first 4 groups of Figure 3d are very trivial, albeit with a significant p value. With such small changes, I am not sure what the biological significance is? Maybe loss-of-function experiments can show a clearer picture?

Our Reply: Thanks to the reviewer's expert comment and suggestion to use C666-1 cell line instead of HK1-EBV as C666-1 does not express endogenous CYLD protein. As shown in the new Figure 5e with C666-1 cells, we found that the inhibitory activity of CYLD on NF-kB was highly significant (reduced by 37.2%; $P<0.0001$) and CYLD mutants S600F and AL527fs were not able to suppress NF-kB activity in the reporter assay, thus confirming the loss-of-function nature of these mutants in C666-1 cell growth (new Figure 5c) and NF-kB activity (new Figure 5e). The less dramatic results previously observed with the CYLD mutants in HK1-EBV cell background could be due to the presence of endogenous CYLD

WT protein which may impose some functional interference on the CYLD mutants (such as competition for binding partners or CYLD interacting proteins, etc). Thus, we are grateful for the reviewer's suggestion to use C666-1, a cleaner cell model for the functional studies in relation to NF- κ B aberrations in NPC (new Figure 5c-e in this revised manuscript). Loss-of-function experiment was not relevant for C666-1 cells since this cell line harbors an endogenous *CYLD* frameshift mutation resulting in the loss of CYLD protein expression. The respective text has been updated (p.11-12 of this revised manuscript).

Comment 7: The data suggests that CYLD can only suppress NF-KB activity in the absence but not the presence of TNF-a treatment (Figure 3d)? What might be the reason?

Our Reply: We appreciate reviewer's expert comment and suggestion to use C666-1 cell line instead of HK1-EBV, as C666-1 does not express endogenous CYLD protein. Our findings in C666-1 indicate that significant suppression of NF- κ B activity was noted even in the absence of any TNF-alpha manipulation (since C666-1 cells have constitutively activated NF- κ B signaling, which is a more relevant biological model for NPC). Please see the new Figure 5e in this revised manuscript.

Comment 8: CYLD, S600F and AL527fs each seem to have different roles in the absence of TNF-a treatment, but the same role in the presence of TNF-a treatment (Figure 3d)?

Our Reply: As addressed above, we agree with the reviewer that the HK1-EBV cell model may not be an appropriate model to study the function of *CYLD* mutants due to the possible interference by endogenous CYLD WT protein in that cell line which may give rise to less dramatic functional results on the mutant. Per the reviewer's suggestion, we have repeated all of the experiments using C666-1 which does not express endogenous CYLD protein and has constitutively activated NF- κ B signaling (Chung GT *et al.* J Pathol. 2013 Oct;231(2):158-67), thus does not require any external artificial TNF-alpha stimulation. It is important to note that, significant suppression of endogenous NF- κ B activity was evident (~40%, N=19, P<0.0001) upon *CYLD* WT expression in C666-1 cells (which is highly consistent with the extent of growth inhibition observed (new Figure 5c) and the *CYLD* mutants S600F and AL527fs were not able to suppress NF- κ B activity in the reporter assay. Suppression of endogenous NF- κ B activity in C666-1 cells (as suggested by the reviewer) did not require any external artificial stimulation of NF- κ B activity as C666-1 cells have constitutively activated NF- κ B signaling.

Comment 9: Survival data of the patients should be provided in the manuscript.

Our Reply: We have revised our Supplementary Dataset to include the patients' original survival data and information (Supplementary Dataset table "NPC Clinicopathological details").

Reviewer #2

Comment 1: I don't see a table listing all the somatic mutations. This will be essential for the community to have this information reported in a supplemental table. It should also include the gene, protein altering event if any, protein mutations caused, variant allele frequency, whether its been reported in COSMIC etc.

Our Reply: Thank you for your comments. We apologize that the list of mutation has been included as Supplementary Dataset (WES NPC MAF (N=111) file) instead of a Supplementary Table (in a powerpoint slide) primarily due to the large file size. Please be informed that we have also deposited these mutational data to dbGAP-NHGRI (Study ID: 20055, Nasopharynx Cancer Whole Exome Sequencing) and European Nucleotide Archive (ENA) (Accession number PRJEB12830, Whole-genome sequencing of matched normal and tumor samples of nasopharyngeal carcinoma patients). As suggested by the reviewer, the respective allele frequencies of each mutation, "# of identical mutations in reported in COSMIC" and "Mutated gene Reported in COSMIC" columns have been added in the "WES NPC MAF (N=111) file" file in this revised version.

Comment 2: Any rare germline mutations for the genes highlighted in the paper should also be listed in a supplemental table.

Our Reply: As per the reviewer's request, we looked within germline calls in matched tumor-

normal samples for cancer-associated genes in our study. We gathered all the variants that were rejected by Mutect algorithm, in part or in full, due to the variant allele being present in the matched normal. A total of 89 potential germline variants were identified. They are found in genes highlighted in the paper or are present in the COSMIC database (in at least 3 samples). We mapped these to the Exome Aggregation Consortium (ExAC) population database containing exome data for over 60,000 individuals (Lek et al, Nature 2015, 10.1038/nature19057), and dbSNP build 134, and Clinvar 12.03.20 using Oncotator v1.9.0.0 (Ramos et al, Hum Mutat 2015 10.1002/humu.22771). Of the 89 variants, 44 were likely polymorphisms with observed allele frequencies (AFs) of at least 0.1%, 15 variants had low AFs (<0.1%) and 30 were not present in any of these databases. These variants did not correspond to any known pathogenicity variants as found in ClinVar. We have included these variants in the supplemental data in the "Germline variants identified in NPC patients" file.

Comment 3: It was interesting that mutation burden was prognostic of survival but there was no correlation with other clinical factors. A Cox proportional hazard model should also be built using the mutational burden data along with the other clinical factors to evaluate the prognostic value of these factors.

Our Reply: We thank the reviewer for the insightful suggestion. We have done this as follows: first we performed univariate analysis by testing each clinical factor (number of SNVs, age, sex, family history, smoking, drinking, histological types, tumor stage, local recurrence, metastases at diagnosis) on their own using Kaplan-Meier survival curves for significance. We then checked if the proportional hazards assumption was met for these factors (using the `cox.zph` function) and none of the factors significantly violated the assumption.

Finally, we fit the significant factors into a coxPH model, using the R package 'survival,' and using a robust method estimate of standard error. There were 65 primary tumors for which all these clinical data were complete. As seen below, the mutational burden covariate had the most significant coefficient, and the only other covariate of significance is age. We did fit a model to see if there was any interaction between mutational burden and age, but did not find any significance. Thus age does affect survival, but the dominant hazard in our dataset is mutational burden.

Factors	coef	exp(coef)	robust SE	z	P value
No. of SNVs (high/mid vs low)	1.86E+01	1.19E+08	5.48E-01	33.94	2.00E-16
metastases at diagnosis	2.26E-01	1.25E+00	7.25E-01	0.31	0.755
Local Recurrent	1.19E+00	3.30E+00	6.40E-01	1.86	0.063
drinking history	7.52E-01	2.12E+00	4.98E-01	1.51	0.131
age	6.93E-02	1.07E+00	1.76E-02	3.93	8.60E-05
smoking history	1.06E+00	2.88E+00	5.46E-01	1.94	0.052

Comment 4: The methods section for mutational signatures needs to be expanded. The SomaticSignatures package was used but the details of what was done are not given. Was NMF used to derive 30 signatures or was another number used? Was the number of signatures to use assessed? Which data (beyond this data set) was included to derive the signatures? How were the underlying causes (e.g. mismatch repair) of these signature applied, manual observation and comparison?

Our Reply: We used the SomaticSignatures (Gehring et al, Bioinformatics 2015 Nov 15;31(22):3673-5) package to first calculate an overall mutation signature across the 70 primaries dataset. We then used NMF clustering to fit the data into two to ten signatures (shown below). We found that four signatures described the majority of the variance within the samples, with the smallest error range, and that increasing the number of signatures to five and beyond did not strongly improve the approximation of the data.

We correlated our signatures to known mutational processes from the COSMIC database (version 8.12.2015.txt) using Pearson's correlation coefficient. Such a table is shown below for the set of 70 primary samples (added as a "COSMIC Signature Analysis" table in the Supplementary Dataset), where it can be seen that the S1 signature correlates best to COSMIC signature C1 (5-methylcytosine deamination), S2 to C2 and C13 (AID/APOBEC), and S3 and S4 to C6/C15 (mismatch repair). Manual comparison with the COSMIC signature plots also supported C1, C2/C13, and C6/C15 as the cancer-associated signatures that best approximate our identified signatures.

S1-S4: NMF-identified signatures

C1-C30: COSMIC cancer signatures

Correlation values: greyed out if under 0.5

COSMIC signature	S1	S2	S3	S4	COSMIC signature description
C1	0.86	0.25	0.51	0.70	result of endogenous mutational proce
C2	0.04	0.83	0.22	0.18	has been attributed to activity of the AI
C13	-0.04	0.67	-0.02	-0.08	AID/APOBEC. occurs with C2
C3	-0.23	0.14	-0.06	-0.05	failure of DNA double-strand break-rep
C4	-0.10	-0.07	0.12	-0.15	smoking, tobacco mutagens. head and r
C5	0.11	0.09	0.33	0.46	all cancer types, unknown mechanism
C6	0.62	0.14	0.64	0.59	defective DNA mismatch repair and is fc
C20	0.27	0.01	0.27	0.34	defective DNA mismatch repair
C26	0.06	0.01	0.13	0.29	defective DNA mismatch repair
C15	0.19	0.13	0.62	0.45	defective DNA mismatch repair
C14	0.25	0.15	0.45	0.45	unknown, but always (>200 somatic mu
C7	0.17	0.51	0.34	0.34	UV light exposure
C8	0.01	-0.04	0.08	-0.01	unknown
C9	-0.04	-0.08	0.03	0.05	polymerase N, activity of AID during son
C10	0.29	0.28	0.14	0.13	associated with POLE, maybe ultrahypei
C11	0.09	0.29	0.31	0.32	alkylating agent treatment
C12	-0.10	-0.05	0.07	0.21	unknown
C16	-0.14	0.10	0.10	0.16	unknown
C17	0.01	-0.01	0.10	0.05	unknown
C18	0.01	0.06	0.24	-0.08	unknown
C19	0.23	0.19	0.35	0.46	unknown
C21	0.11	-0.05	0.08	0.21	unknown
C22	-0.06	-0.13	-0.13	-0.11	exposure to aristolochic acid
C23	0.09	0.10	0.27	0.29	unknown
C24	-0.07	0.07	0.14	-0.08	exposure to aflatoxin
C25	0.08	0.05	0.00	0.14	unknown
C27	-0.02	-0.02	-0.03	-0.04	unknown
C28	-0.06	-0.07	-0.05	-0.08	unknown
C29	0.13	0.01	0.22	0.04	tobacco chewing habit
C30	0.14	0.49	0.37	0.44	unknown

Thus, three known cancer signatures most correlated with our primary NPC dataset: 5-methylcytosine deamination, AID/APOBEC, and mismatch repair. As we increased the number of signatures assessed using the NMF method, we did not identify any additional highly correlated COSMIC signatures.

We have updated the manuscript text to clarify our methods on p.17, and have also included the 4 signatures we identified in numerical format in the Supplementary Data as the “COSMIC Signature Analysis” table.

Comment 5: Was TP53 mutation status (or any of the other significantly mutated genes) prognostic of survival?

Our Reply: In our cohort of primary NPCs, we found that *TP53* mutation status was not correlated with the disease-free survival ($p=0.82$) and overall survival ($P=0.42$) (Log-Rank (Mantel-Cox) test). The text has been modified on p.7.

Comment 6: As the majority of CYLD mutations are in the DUB domain, the authors should check the expression status and stability of CYLD substrates (RIP1, TRAF2, BCL3) on endogenous level.

Our Reply: Thank you for the suggestion. We have analyzed the effects of *CYLD* mutants and *CYLD*WT on RIP1, TRAF2 and Bcl-3 in C666-1 cells (a cell line suggested by Reviewer 1). Changes in Bcl-3 nuclear translocation was observed, but there is no significant change in endogenous total RIP1 and TRAF2 protein levels upon *CYLD* WT or mutant expression in C666-1 cells (please see “Figure for Reviewer Only” as provided below). We have modified the text accordingly: Nuclear translocation of Bcl-3, a well-known step for activation of NPC specific NF- κ B signal, p50/p50/BCL3, was inhibited upon *CYLD* WT expression, but not by *CYLD* mutants (Supplementary Fig. S12a, also p.12 of the text). We further demonstrated that only WT *CYLD*, but not *CYLD* mutants showed the DUB activity in Bcl3 (Supplementary Fig. S12b). It appears that inhibition of Bcl3 nuclear translocation is the major mechanism of *CYLD* WT in regulating distinct NF- κ B signal in C666-1 cells.

Figure for Reviewer Only: *CYLD* mutations did not alter TRAF2 nor RIP1 levels in C666-1 cells. C666-1 cells were transfected with *EGFP-Ctrl*, *CYLD* WT, and *CYLD* mutants for 48 hours. Western blot analysis showed unchanged levels of TRAF2 and RIP1 proteins by *CYLD* WT or *CYLD* mutants. Graphs below showing cumulative results from 4 independent experiments.

Comment 7: A lot of mutations in TRAF3 are cysteine mutants, which could affect TRAF3 protein expression and stability. The authors should also check p100 (alternative NF- κ B) processing, which is affected by TRAF3 expression.

Our Reply: Out of the 9 *TRAF3* point mutants, 4 are frameshift mutants and 5 are non frameshifting mutations which mainly affect cysteine residues in the RING domain of TRAF3 (C55G, C76Y, C88Y). Per the suggestions of the reviewer, we generated all these 3 RING domain mutants and examined their effects on p100 processing in NPC cells (HK1). As shown in the figure below, our cumulative results from 3 independent experiments showed that expression of *CYLD* WT protein resulted in an increased ratio of p100/p52 (indicative of less p100 processing) when compared to the *EGFP* control ($P=0.0169$), while the p100/p52 ratios for *TRAF3* C56G, *TRAF3* C76Y and *TRAF3* C88Y were not significantly different from that of the *EGFP*-control ($P=n.s.$), supportive of a loss-of-function activity in these mutants (p.11 of the revised text, Supplementary Fig S9).

Comment 8: Data in figure 3d show minimal effect making it hard to correlate this effect to cell growth effect. Is cell viability affected?

Our Reply: Thanks to the reviewer's expert comment. We believe that this suboptimal effect observed in HK1-EBV cells could be due to the presence of endogenous CYLD WT protein in HK1-EBV, which may mask the effects of the CYLD mutants. Therefore, per suggestion of Reviewer 1 to use another NPC cell line (C666-1) with no detectable endogenous CYLD WT protein, we have repeated this experiment and observed a marked suppression of NF- κ B activity even in the absence of any TNF- α manipulation. This is because C666-1 cells are known to have constitutively activated NF- κ B signaling, which is a more relevant biological model for NPC. Please see the new Figure 5c in this revised manuscript. In C666-1 cells, the significant suppressive effects on NF- κ B signaling and cell growth were noted in CYLD WT, while both mutants lost their effects on NF- κ B suppression (Figures 5c and 5e). The text has been revised on p.11-12.

Comment 9: The authors should examine LMP1 expression by western blotting to support the data in figure 4a, especially for WT protein in high expression cases.

Our Reply: Since majority of our tumor cases were FFPE specimens of endoscopic biopsies with small size, it is not possible for us to perform western blotting on these samples. In our previous study (Chung et al. 2013), we have identified the genetic alterations of *TRAF3*, *NFKBIA* and other *NFKB* related genes in a panel of NPC cell line (C666-1) and patient derived xenografts (PDXs) (xeno-2117, xeno-1915, xeno-99186, C17, xeno-666) which show an absence or low LMP1 expression. In a PDX with high LMP1 expression (C15), no genetic alteration of any NF- κ B related genes was detected. A figure of the Western blot analysis of LMP1 expression in these NPC cell line/PDXs is shown below (for reviewer's reference only). The somatic alterations of NF- κ B related genes in these tumors are also listed. LMP1 was highly expressed in the PDX xeno-C15 in which no genetic alteration of NF- κ B related genes was found.

NPC cell line & PDXs	LMP1 expression	Genetic alterations of NF-κB related genes
C666-1	-ve	TRAF3 and CYLD frameshift mutations
Xeno-666	-ve	TRAF3 and CYLD frameshift mutations
Xeno-2117	-ve	LTBR amplification
Xeno-99286	-ve	TRAF3 frameshift mutation
Xeno-C17	-ve	NFKB1A homozygous deletion
Xeno-C15	+ve	None

Comment 10: Please increase the font size for text in the figures - it is barely legible.

Our Reply: Thanks for the suggestion and we have increased the font size to a minimal of font 8 or larger for clarity purposes. We have also expanded the size of some figures and panels, thus move some figure subpanels into new bigger figures for clarity purposes.

Reviewer #3:

Comment 1: All signatures derived from the non-negative matrix factorization (NMF)

analysis should be shown in Supp. Figures.

Our Reply: As per the reviewer's suggestion, we have now included numeric tables of the signatures derived from NMF analysis for the primary and recurrence sample sets (added as a "COSMIC Signature Analysis" table in the Supplementary Dataset).

Comment 2: Figure 2a - please explain what the asterisks stand for. * In 2 tumors the authors detected 2051 and 817 somatic coding mutations, and link the reader to Figure2a. However, the axis Y extends only up to around 500 mutations per sample. The authors have to re-plot the figure or explain this in the legends.

Our Reply: Thank you for your suggestion to improve the accuracy of presentation. We have revised the Figure 2 and removed the *. We have extended the axis Y to show the actual number of mutations in the cases with 2051 and 817 somatic mutations.

Comment 3: Supp. Figure 3 - please explain what the arrows emphasize. The arrows are not mentioned neither in the text nor in the legends.

Our Reply: We apologize for the confusion due to our display on the figure. We have provided all original mutation data for each NPC pair as Supplementary Dataset in this revised manuscript (labeled as the "Mutational events of paired NPCs: primary/recurrent, primary/lymph node metastasis, and metastatic site 1/metastatic site 2." file) and thus removed all arrows in the figure.

Comment 4: In addition to the data shown in Supp. Figure 3, please add a Supplemental Table listing all the mutations shared between the paired primary/recurrent and primary/metastatic NPC. Also, were silent/synonymous mutations considered?

Our Reply: Thank you for the suggestion to improve the clarity of presentation. We have provided all original mutation data (including both silent and non-synonymous mutations) for each NPC pair as Supplementary Dataset in this revised manuscript (labeled as the "Primary-recurrences or mets" file) in the paired primary/recurrent and primary/metastatic NPC pairs.

Comment 5: Page 10 - "one case harboring homozygous deletion confirmed by Fluorescence in situ Hybridization (FISH; Supplementary Fig. S7)". It appears the wrong figure is referenced. FISH is actually shown in Supplementary Figures S8 and S9, not S7. Figure S7 as it appears in the supplemental material is not mentioned in the text. Should be corrected.

Our Reply: We apologize for the unclear sentence and confusions caused. In the revised manuscripts, the sentence was changed as "one case harboring homozygous deletion (Supplementary Fig. S5). The *CYLD* homozygous deletion was further confirmed by Fluorescence in situ Hybridization (FISH) (Supplementary Fig. S7).

Comment 6: It seems that HK1-EBV cells are already expressing relatively high levels of *CYLD*, therefore, it is not a good cell line model for overexpression experiments. The functional experiments shown in Figure 3C should be repeated in a cell line depleted for *CYLD* or at least expressing a low *CYLD* level. For example, the biological role of wt-*CYLD* or S600F *CYLD* can be assessed in C666-1 cells, which do not express *CYLD* protein. It would also be interesting to see if *CYLD* knockdown in HK1-EBV cells will result in the same phenotypical changes as those seen in cells expressing the mutant variants.

Our Reply: We thank the reviewers' (Reviewer 1 and Reviewer 3) expert comment. We fully agree with the reviewers that the C666-1 cell line, which contains a disrupted allele of *CYLD* and depleted for *CYLD* protein. With C666-1's clean background with no *CYLD* protein expression, we repeated the functional experiments (new data shown in Fig 5c-e). We found that the expression of *CYLD* wildtype gene suppressed C666-1 cell proliferation and anchorage-independent growth on soft agar, while expression of patient-derived *CYLD* point and truncating mutations resulted in the loss of these tumor-suppressive activities (Fig. 5c and 5d). This loss-of-function activity of *CYLD* mutants was confirmed in another NPC cell line, HK1-EBV (Supplementary Fig. S11). Further, a higher NF- κ B transcriptional activity (as measured by an NF- κ B-luciferase reporter assay) was observed in C666-1 cells expressing the patient-derived *CYLD* mutants versus wild-type (Fig. 5e). The text has been modified on p.11-12.

We have performed *CYLD* siRNA experiments on both HK1-EBV and C666-1 cells. However, since C666-1 cells do not express any endogenous *CYLD*, there was no observable changes upon *CYLD* siRNA transfection vs. control siRNA (data not shown). For HK1-EBV, we also observed no significant change in cell growth over 72 hours (data not shown). This is likely due to the lack of p50/p50/BCL3 nuclear NF- κ B signals in this EBV-

reinfected NPC cells. Furthermore, as shown in Supplementary Figure S11, *CYLD* WT expression in C666-1 cells (which is a more relevant cell line for *CYLD* functional study as suggested by the reviewer) can inhibit nuclear BCL3 accumulation, which is known to be important for its growth-promoting effects.

Our results indicate that *CYLD* mutants are functionally distinct from *CYLD* knockdown, suggesting the *CYLD* mutants are likely to be exerting some functional effects or dominant negative effects (not shared by simple knockdown) which contribute to C666-1 cell proliferation and anchorage-independent growth phenotypes observed (demonstrated in C666-1 cells as suggested by the reviewer). Dominant negative effects of *CYLD* mutant have been suggested previously (Miliani de Marval P, et al, Cancer Prev Res (Phila) 2011 Jun; 4(6): 851-9).

Comment 7: Mapping of the altered genes to pathways is an important characteristic of the study but the main text provides little information. It is not clear what software or model the authors used for pathways analysis. The methods section entitled "Pathway analysis" lacks any explanation about the pathways analysis and has to be significantly expended. There is little statistical support given for any sort of enrichment analyses described in the text.

Our Reply: The reviewer is correct in that this was not a rigorous pathway analysis, and rather more of a summary figure. This figure was manually constructed starting with well known signaling pathways (i.e. PI3K/MTOR, cell cycle, TRAF signaling pathways), supplemented by literature curation, and highlighting genes that were recurrently altered in our data set. TCGA studies have frequently employed this type of approach (i.e. Cancer Genome Atlas Network. Nature. 2015 Jan 29;517(7536):576-82) to summarize recurrently altered pathways in the dataset. We did utilize pathway analysis programs such as DAVID or IPA on our significantly mutated genesets; however, we did not find any additional insights.

We thank the reviewer for pointing out this confusion. As such, we have renamed this section "Pathway diagram" which we hope better clarifies the text and the method. The text has been modified on p.14.

Comment 8: Was there any orthogonal validation done on any of the variants detected?

Our Reply: To call somatic mutations in the original manuscript we used MuTect, an established algorithm developed at the Broad Institute of Harvard and MIT optimized for

mutation calling in stromally contaminated tumor tissues, including FFPE samples, with high validation rates reported in various studies to date including the original report of Cibulskis *et al.* (Nat Biotechnol. 2013 Mar;31(3):213-9) as well as Van Allen *et al* (Nat Medicine, 2014 Jun; 20(6): 682–688) which specifically addressed the use of this method in FFPE samples. While we are not aware of a consensus method for orthogonal validation of WES mutation calls from stromally admixed FFPE samples given the challenge of matching the sensitivity of hybrid capture, especially for mutation calls at lower allele fractions, we have now added additional validation of mutation calls using the Fluidigm method.

Specifically, we used the Fluidigm Access Array microfluidic system, which allows for multiplexed, high-throughput, amplicon sequencing using custom targeted primers. We focused on potentially cancer-related somatic mutations that were either discussed in the manuscript or seen in at least three samples in the COSMIC database, and all called somatic variants in trios. We designed primers for a total of 464 positions on 2 Fluidigm chips. Raw FASTQ result files were processed and aligned using bowtie2 on standard settings. Overall, we had an average of 216,649 +/- 48,880 reads, 85% +/- 7% alignment rate and coverage of 660X +/- 181 across the 74 samples.

We manually reviewed variants using IGV and found that we validated 82% (242/294) of variants, including both SNVs and indels in the cases with minimum original allele frequencies of 15% and 71% of mutations (264/373) at a 5% allele fraction cut point (table below). We felt it was reasonable to see that the validation rate increased, and the number of uncovered positions decreased as the original allele frequency of detected variants increased. More specifically for the trios (matched normal also sequenced), we were able to validate the presence of similar shared variants from WES in HKNPC012, HKNPC008 and HKNPC009. These results recapitulate what we observed in our original manuscript.

minimum original AF	# validated in tumor validation sample (and not in normal, where available)	#not validated (not present in validation tumor sample)	not covered in one of the validation samples	validation %
15%	242	52	50	82%
10%	298	85	67	78%
5%	264	109	77	71%

Comment 9: For the variants called, the authors should provide read counts for non-synonymous mutant and reference for each relevant lesion at that position (as a Supplemental excel Table). As it stands, there is only information given about average coverage of 86X. This will be useful in understanding the quality of the data being presented.

Our Reply: Thank you for the suggestion. All variant read counts and the respective reference read counts have been added for all mutations in the Supplementary Data as the “WES NPC MAF (N=111)” file in this revised manuscript.

Comment 10: Information sharing strategies are not discussed in this paper and should be added.

Our Reply: We have deposited our WES data to dbGAP-NHGRI (Study ID: 20055, Nasopharynx Cancer Whole Exome Sequencing) and European Nucleotide Archive (ENA) (Accession number PRJEB12830, Whole-genome sequencing of matched normal and tumor samples of nasopharyngeal carcinoma patients). The text has been updated on p.16-18.

Comment 11: Mutational nomenclature should be in standard format, that is c. and p.

Our Reply: Thank you for the comment. We have fixed the related mutational data with this nomenclature (Supplementary Dataset tables “WES NPC MAF (N=111)” “Primary_Recurrences or mets”, “Validation of sequence variants” files), as well as all figures including Figure 3, Supplementary Figures S1, S9, S10, S11, and S12).

Comment 12: Other than EBV status, there are no patient characteristics described that could also relate to the disease and the mutational burden. For example, were these smokers? Alcohol intake? If this information is available it should be added.

Our Reply: Smoking and drinking history has been added onto Supplementary Data table “NPC Clinicopathological Details” file in this revised manuscript.

Comment 13: The manuscript should be formatted in accordance to the journal's requirements. The abstract is missing and should be added. The titles for new paragraph sections should be added. A formal discussion section should be added.

Our Reply: We have edited the text for Nature Communications format. We have included a formal introduction, subtitles, and included a Discussion section. Due to the fact that Nat.

Comms. Allows a maximum of 10 figures in the manuscript, we have increased the size of our figures and expanded to a total of 8 figures for legibility as commented by Reviewer #2 to increase the font size within the figures.

REVIEWERS' COMMENTS:

Reviewer #1 (Remarks to the Author):

The authors have answered some of the most important critiques with clear results from their complementary experiments. However, a few questions remain to be addressed as outlined below:

In the clonality analysis (Supplementary Figures S2-S3), the authors used variant allele frequency to determine clonality, which is not accurate. This is because variant allele frequency can be influenced by the ploidy, tumor cellularity as well as copy number alterations. The authors should instead determine cancer cell fraction of the variants by integrative analysis of both variant allele frequency, tumor cellularity and copy number changes.

In the experiment testing TRAF3's activity (Supplementary Figure S9), the quantification plot of p100 /p52 ratio was inconsistent with what could be interpreted in Western blot results. Specifically, in Western blot, all mutants showed clear decreased p100 /p52 ratio compared with the EGFP control group. But in the quantification data, they had almost exact the same ratio. This discrepancy needs to be clarified, as it may lead to a revised conclusion that mutant TRAF3 could further inhibit the function of endogenous TRAF3 (i.e., a dominant negative role?).

Some of the descriptions in the manuscript need to be revised in order to become more scientifically accurate. For example, it was written "It can be envisioned that clinically approved NF- κ B inhibitors, Bortezomib and a newly developed small molecule inhibitor targeted Bcl3, JS6 can potentially be used as novel therapeutics for NPC patients." I am not sure whether Bortezomib (a well-known proteasome inhibitor) has an established activity against NF- κ B. At the minimum, the authors should cite references to support this statement. Likewise, appropriate references should be cited for JS6.

The authors should tone down some of the statements. For example, it was described "Our WES (from FFPE tissues) and WGS (from fresh frozen tissues) gene copy changes were highly consistent." However, some inconsistencies can be readily seen, e.g., Chr1q was frequently gained in WES but not WGS; similarly, Chr3q has more gains in WES than WGS. Also, in line 103, "the vast majority of NPCs display activation of the NF- κ B signaling pathway as a result of somatic inactivating mutations in negative regulators of NF- κ B", but Figure 6 shows less than half of cases were affected by the endogenous genetic lesions targeting NF- κ B pathway.

The manuscript should be carefully proofread by additional native English-speaking

scientists, as a number of grammar errors were found, such as: 1) “Losses of 14q and 16q, where multiple negative regulators of NF- κ B pathway are located [NFKBIA (14q13), TRAF3 (14q32.3), CYLD (16q12.1), NLRC5 (16q13)].” 2) Line 356; 3) Line 210, etc.

Reviewer #2 (Remarks to the Author):

No further comments.

Reviewer #3 (Remarks to the Author):

The authors attempted to respond to all my previous comments and have went to a great length to address my concerns by performing additional experiments and analyses. After giving this paper a fresh look I believe that this publication will be of interest to the broad clinical and scientific audience.

Minor comments:

* Line 123: The sentence “To identify the mutagenic processes operative” is missing “if” or “whether” after “identify”.

* Line 138: coma after NF- κ B pathway genes.

* Line 198: “.....only TP53 mutations were the only shared annotated cancer gene as defined by the Cancer Gene Census, COSMIC.” This part of the sentence is confusing and have to be rephrases.

* Line 203: first “of” should be removed.

* Line 215: Sentence starts with “Losses” should be rephrased.

* Abbreviation for FISH should be brought to line 211, where it is first mentioned.

: Line 232: first “of” should be substituted with “in”.

* Line 242: remove “Indeed”.

* Line 285: substitute “among” with “between”.

Point-by-Point Reply:

We would like to thank the reviewers and editor for their thorough reading of our manuscript as well as for their invaluable comments. We have followed their comments closely and feel that their suggestions have indeed further strengthened our manuscript. Below are our responses.

Reviewer #1:

Comment 1: In the clonality analysis (Supplementary Figures S2-S3), the authors used variant allele frequency to determine clonality, which is not accurate. This is because variant allele frequency can be influenced by the ploidy, tumor cellularity as well as copy number alterations. The authors should instead determine cancer cell fraction of the variants by integrative analysis of both variant allele frequency, tumor cellularity and copy number changes.

Our Reply: Thank you very much for the reviewer's suggestion. We agree that all of these factors can affect the detected variant allele frequencies (VAFs). In Supplementary Figures 2-3, we had chosen to use simple clonality graphs to compare the overall similarity/dissimilarity of SNV clonality patterns in tumors from the same patient. Following the above comment we have improved this analysis by filtering out SNVs that occur in regions of copy number gain or loss in either tumor sample. This allowed us to remove the complicating effects of copy number alterations on VAFs and resulted in slightly cleaner scatter plots. This is most obvious for trio 9.

To account for the limited number of trios in our analysis we have revised the result section in page 10 as following: "For these NPC patients, the recurrent or metastatic lesion appeared to show a SNV clonality distribution different from the primary tumor, supportive of emerging new subclone(s) in recurrences. On the other hand, the liver metastases from patients HKNPC-008 and HKNPC-009 both showed a relatively higher percentage of shared mutations among two distinct regions of liver metastases (Supplementary Fig. 3; Supplementary Data)."

Comment 2: In the experiment testing TRAF3's activity (Supplementary Figure S9), the

quantification plot of p100 /p52 ratio was inconsistent with what could be interpreted in Western blot results. Specifically, in Western blot, all mutants showed clear decreased p100 /p52 ratio compared with the EGFP control group. But in the quantification data, they had almost exact the same ratio. This discrepancy needs to be clarified, as it may lead to a revised conclusion that mutant TRAF3 could further inhibit the function of endogenous TRAF3 (i.e., a dominant negative role?).

Our Reply: In the Supplementary Figure 9, the cumulative plot of quantification of p100 /p52 levels was derived from three independent experiments after normalization with actin expression level. Despite varied levels of p100 among these independent experiments (please see graph below, as indicated by the SD), our results from 3 independent experiments demonstrated that the p100/p52 ratios in the TRAF3 mutants and EGFP control groups are clearly lower than that in the wild type TRAF3 transfected cells (Supplementary Fig. 9). The findings support the loss-of-function activities of the TRAF3 mutants.

Comment 3: Some of the descriptions in the manuscript need to be revised in order to become more scientifically accurate. For example, it was written “It can be envisioned that clinically approved NF- κ B inhibitors, Bortezomib and a newly developed small molecule inhibitor targeted Bcl3, JS6 can potentially be used as novel therapeutics for NPC patients.” I am not sure whether Bortezomib (a well-known proteasome inhibitor) has an established activity against NF- κ B. At the minimum, the authors should cite references to support this statement. Likewise, appropriate references should be cited for JS6.

Our Reply: In page 15 of our revised manuscript, the sentence was revised as “It could be envisioned that NF- κ B inhibitors and the newly developed small molecule inhibitors targeting Bcl3 might be used as novel therapeutics for NPC patients^{25,26}.” Two new references (ref. 25 and ref. 26) have been added for the role of bortezomib and other inhibitors in targeting NF- κ B pathways in human cancer and the recent developed BCL3 inhibitors.

Comment 4: The authors should tone down some of the statements. For example, it was described “Our WES (from FFPE tissues) and WGS (from fresh frozen tissues) gene copy changes were highly consistent.” However, some inconsistencies can be readily seen, e.g., Chr1q was frequently gained in WES but not WGS; similarly, Chr3q has more gains in WES than WGS. Also, in line 103, “the vast majority of NPCs display activation of the NF- κ B signaling pathway as a result of somatic inactivating mutations in negative regulators of NF- κ B”, but Figure 6 shows less than half of cases were affected by the endogenous genetic lesions targeting NF- κ B pathway.

Our Reply: We have toned down some of the statements according to reviewer’s suggestions. In the revised manuscript, the sentences were changed to “Similar gene copy changes were observed in our WES (from FFPE tissues) and WGS (from fresh frozen tissues) studies.” (page 10) and “.....reveals that majority of NPCs display activation of the NF- κ B signaling pathway as a result of somatic inactivating mutations in negative regulators of NF- κ B.” (page 5).

Comment 5: The manuscript should be carefully proofread by additional native English-speaking scientists, as a number of grammar errors were found, such as: 1) “Losses of 14q and 16q, where multiple negative regulators of NF- κ B pathway are located [NFKBIA (14q13), TRAF3 (14q32.3), CYLD (16q12.1), NLRC5 (16q13)].” 2) Line 356; 3) Line 210, etc.

Our Reply: We have carefully proofread the revised manuscript. The sentence mentioned was rephrased as “Losses of 14q and 16q may serve as one of the mechanisms for inactivating multiple negative regulators of NF- κ B pathway, such as *NFKBIA* (14q13), *TRAF3* (14q32.3), *CYLD* (16q12.1), and *NLRC5* (16q13).” The sentences in Line 356 and Line 210 were revised and shown on page 16 and page 10 of our revised manuscript, respectively.

Reviewer #2

Comment: No further comments

Our Reply: Thank you very much for the reviewer's support.

Reviewer #3:

Comment 1: The authors attempted to respond to all my previous comments and have went to a great length to address my concerns by performing additional experiments and analyses. After giving this paper a fresh look I believe that this publication will be of interest to the broad clinical and scientific audience.

Our Reply: We thank the reviewers' for their positive comments on the study.

Comment 2: Line 123: The sentence "To identify the mutagenic processes operative" is missing "if" or "whether" after "identify".

Our Reply: We have added the word "whether" in the sentence as reviewer's suggestion in Page 6 of the revised manuscript.

Comment 3: Line 198: ".....only TP53 mutations were the only shared annotated cancer gene as defined by the Cancer Gene Census, COSMIC." This part of the sentence is confusing and have to be rephrases.

Our Reply: In page 10 of our revised manuscript, the sentence was rephrased as "In the paired primary/recurrent HNSCC samples, we also found that *TP53* was the only shared

annotated mutated cancer gene as defined by the Cancer Gene Census, COSMIC.”

Comment 4: Line 203: first “of” should be removed.

Our Reply: The word “of” was removed from the sentence in page 10 of the revised manuscript.

Comment 5: Line 215: Sentence starts with “Losses” should be rephrased.

Our Reply: In page 10 of our revised manuscript, the sentence was rephrased as “Losses of 14q and 16q may serve as one of the mechanisms for inactivating multiple negative regulators of NF- κ B pathway, such as *NFKBIA* (14q13), *TRAF3* (14q32.3), *CYLD* (16q12.1), and *NLRC5* (16q13).”

Comment 6: Abbreviation for FISH should be brought to line 211, where it is first mentioned.

Our Reply: In page 10, the abbreviation for FISH was brought to the sentence “Selected CNVs were further validated and confirmed by Fluorescence in situ Hybridization (FISH) analysis” according to the reviewer’s suggestion.

Comment 7: Line 232: first “of” should be substituted with “in”.

Our Reply: The word “of” in the sentence was substituted with “in” in page 11.

Comment 8: Line 242: remove “Indeed”.

Our Reply: The word “Indeed” in page 12 was deleted as reviewer’s suggestion.

Comment 9: Line 285: substitute “among” with “between”.

Our Reply: In page 13 of our revised manuscript, the sentence was changed to “....that there was an association between PI3K activating events and poor outcome among recurrent and metastatic NPC patients”.